# Dissecting reversible and irreversible single cell state transitions from gene regulatory networks

Daniel A Ramirez [1,2] & Mingyang Lu [1,2]✉

## Abstract

**Understanding cell state transitions and their governing regulatory mechanisms remains one of the fundamental questions in biology. We develop a computational method, state transition inference using cross-cell correlations (STICCC), for predicting reversible and irreversible cell state transitions at single-cell resolution by using gene expression data and a set of gene regulatory interactions. The method is inspired by the fact that the gene expression time delays between regulators and targets can be exploited to infer past and future gene expression states. From applications to both simulated and experimental single-cell gene expression data, we show that STICCC-inferred vector fields capture basins of attraction and irreversible fluxes. By connecting regulatory information with systems' dynamical behaviors, STICCC reveals how network interactions influence reversible and irreversible state transitions. Compared to existing methods that infer pseudotime and RNA velocity, STICCC provides complementary insights into the gene regulation of cell state transitions.**

**Keywords** Gene Regulatory Network (GRN); Systems Biology; Single Cell RNA-seq; RNA Velocity; Cell State Transition
**Subject Categories** Chromatin, Transcription & Genomics; Computational Biology; Methods & Resources

## Introduction

A key question in biology is how gene regulatory networks (GRNs) control biological processes and how this knowledge can shed light on therapeutic strategies for diseases (Gerstein et al, 2012; Schwikowski et al, 2000). A central focus in recent years are those gene networks that control cell state transitions during healthy developmental processes, such as cell differentiation (Okawa et al, 2016) and cell cycle progression (Haase and Wittenberg, 2014), and during disease development, such as tumorigenesis (Chen et al, 2020) and fibrosis (Forte et al, 2020). In many cases, cell state transitions are irreversible due to feedback control or epigenetic mechanisms (Blanco et al, 2021); for example, in the cell cycle, irreversible transitions were found to be achieved by a negative feedback loop creating a one-way toggle switch

(Verdugo et al, 2013). In some other systems, cells can interconvert between distinct stable states stochastically or in response to specific stimuli, such as when certain differentiated cell types dedifferentiate in response to tissue damage (Hormoz et al, 2016; Nichols et al, 2020). These mechanisms may be combined to permit complex decision-making between multiple cell fates according to environmental cues (Doncic and Skotheim, 2013; Tian et al, 2013). Single-cell transcriptomics has become a popular technology to identify distinct transcriptional cell states (Ke et al, 2022), paths of transitions between states (Zhou et al, 2021), and cell-to-cell variations within each state (Yeo et al, 2020). However, it remains challenging to reveal complex patterns of cell state transitions and their underlying regulatory mechanisms. A recent computational approach, named RNA velocity, estimates the instantaneous rate of change of gene expression for cells using single-cell RNA sequencing (scRNA-seq) data (La Manno et al, 2018). In the RNA velocity method, a simple kinetic model is established for each gene to describe RNA processing from the unspliced form to the spliced form, using measured abundances from scRNA-seq data. Because of the time delays during RNA processing, RNA velocity can exploit the memory present in the read counts to predict changes in gene expression. RNA velocity has been demonstrated to capture cell state transitions in several applications (Kimmel et al, 2020; Wolfien et al, 2020; Qiao and Huang, 2021) and has been generalized to consider the temporal sequences of other regulatory events during gene regulation (Bergen et al, 2020; Haensel et al, 2020). This method has also been used to infer pseudotime trajectories during cell differentiation and to infer gene regulatory interactions (Bocci et al, 2022; Lange et al, 2022; Qiu et al, 2022). The ability of RNA velocity to infer gene expression dynamics from static data has made it a powerful tool in single-cell transcriptomics. However, there are a few limitations of existing methods based on RNA velocity. First, these methods only consider the regulatory events of individual genes and fall short in characterizing the relationship between cell state transitions and GRNs. Second, from a dynamical-systems view, state transitions could be reversible and irreversible; however, such a feature of state transitions has rarely been captured in a typical RNA velocity analysis.

To address these limitations, here we develop a new computational tool, named state transition inference using cross-cell correlation (STICCC), to predict the irreversible and reversible directions of state transitions at single-cell resolution using gene expression data and a set of regulator–target interactions (Fig. 1A). This approach assumes that changes in regulators' activity/expression should precede changes in targets' expression—and

[1]Center for Theoretical Biological Physics, Northeastern University, Boston, MA 02115, USA. [2]Department of Bioengineering, Northeastern University, Boston, MA 02115, USA. ✉E-mail: m.lu@northeastern.edu

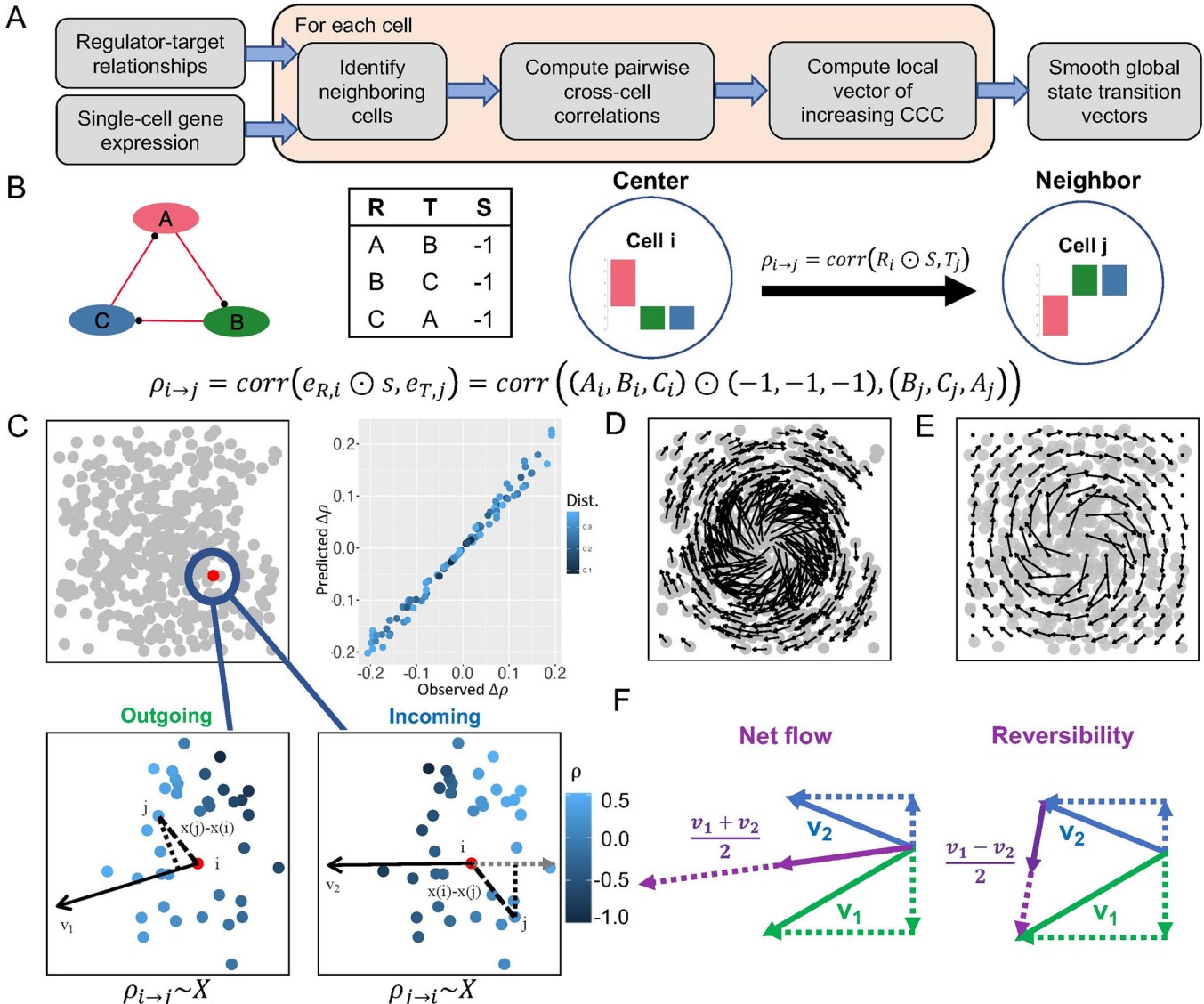

**Figure 1. Overview of the method STICCC.**

(**A**) Workflow of a new algorithm to infer single-cell state transition. The method takes single-cell gene expression data and a set of regulator–target relationships (or topology of a transcriptional regulatory network) as the input. For each cell, STICCC predicts a vector pointing towards its future state by using cross-cell correlations (CCCs). Finally, local smoothing is applied to generate global state transition vectors. (**B**) CCC between a center cell i and its neighboring (proximal in gene expression) cell j, $\rho_{i \to j}$, is defined as the Pearson correlation of regulator expression in cell i with corresponding target expression in cell j. The regulator expression is inverted for inhibitory edges to preserve the sign of the correlation. Here, a repressilator (REP) gene circuit is illustrated as an example. The panel shows the circuit diagram, circuit topology (R regulators, T targets, S regulation signs), and the definition of CCC. (**C**) Illustration of the inference of outgoing transition vectors by STICCC. The top left panel illustrates a scatter plot of single-cell gene expression in a low-dimensional projection. The top right panel shows the relationship between the calculated value of $\rho_{i \to j}$ for each neighboring cell j on the x axis and the predicted value of $\rho_{i \to j}$ based on the linear regression. The bottom panels show a zoomed-in view of the gene expression space surrounding the center cell i (red), with the CCC values from cell i to a neighboring cell j (bottom left) or from each neighboring cell to cell i (bottom right) depicted in a blue gradient. The transition vectors align with the gradient of CCCs, which can be inferred through multiple linear regression. (**D**) Illustration of cell-wise predictions of outgoing transition vectors for a full dataset. (**E**) Smoothed global state transition patterns from (**D**). (**F**) Net flow and reversibility are defined as the average and half the difference of incoming and outgoing vectors, respectively.

therefore, cells' future expression state can be inferred from their current regulatory state. The core of the approach is a metric named cross-cell correlation (CCC), which measures the association between regulator activity/expression in one cell and corresponding target gene expression in a different cell with similar gene expression (Katebi et al, 2020). A high CCC indicates

that the latter cell represents a likely time evolution of the former, according to the provided regulator–target relationships. CCC therefore captures cell transition probabilities using the causal, directional information in the GRN, whereas many diffusion- and manifold-based methods currently in use must leverage additional information, including RNA velocity or predetermined ordering of

cell states, to constrain the directionality of transitions (Lange et al, 2022; Chen et al, 2014). Using the distribution of CCCs from a cell to its neighboring cells ("neighbor" as defined in the gene expression space), one can obtain a vector representing the likely direction of future transition for one cell. Similarly, another transition vector can be inferred using the distribution of CCCs from the neighboring cells to the center cell, indicating the likely direction towards prior states. Using both the incoming and outgoing vectors allows us to decompose the state transitions into the reversible and irreversible components. Compared to related existing techniques, such as pseudotime and RNA velocity, STICCC provides complementary insights by (1) closely integrating regulatory information into the modeling of cell state transitions, (2) emphasizing the local gene expression landscape rather than fitting gene-wise or global kinetic parameters, and (3) explicitly describing both reversible and irreversible state transitions.

In the following, we first summarize the methodology of STICCC, with the details provided in the "Methods" section. We then apply STICCC to several simulated gene expression datasets derived from synthetic gene regulatory circuits. We demonstrate that the algorithm can recapitulate basins of attraction and irreversible flux, both of which can be associated with the dynamical behaviors of the corresponding gene circuits. Inferred transition vectors are robust to noise from dropouts and agree with simulated expression trajectories. Furthermore, we illustrate the application of STICCC on several experimental single-cell gene expression datasets to elucidate diverse patterns of cell state transitions. By comparing state transition vectors from slightly perturbed network topologies, STICCC can also explore how reversible and irreversible state transitions are influenced by regulators and regulatory interactions. STICCC is available as a free R package to the community on GitHub: https://github.com/lusystemsbio/sticcc/.

## Results

### Vector inference of single-cell state transitions

We develop the STICCC method to infer single-cell state transition vectors from single-cell gene expression data and a list of regulator–target gene relationships. STICCC utilizes a metric termed cross-cell correlation (CCC, Eq. (4) in "Methods"), based on delayed correlation (Katebi et al, 2020; Chen et al, 2014; Li et al, 2006), to uncover the transition propensity between two single cells (Fig. 1B). By assuming a linear relationship between the CCCs from a center cell $i$ toward its neighboring cells and the cells' relative gene expression, we can infer the outgoing transition vector $v_1$ of cell $i$ using multiple linear regression (Eqs. (1C) and (5) in "Methods"). The same process can be applied to each cell in a dataset to describe the collective transition pattern (Fig. 1D). For easier visual interpretation, from the cell-wise vectors, we apply inverse distance-weighted smoothing on a uniform grid to create a vector field (Fig. 1E, see "Methods"), which allows to generate a clear representation of the vector field for cell state transitions.

To illustrate the inference of the outgoing transition vectors, we applied it to simulated gene expression data from a repressilator (REP) gene circuit consisting of three genes sequentially inhibiting each other (Fig. 1B) (Elowitz and Leibler, 2000). The gene

expression dynamics of the REP circuit were simulated by a typical ordinary differential equation (ODE) model with Hill kinetics for gene regulation and linear terms for protein degradation (see "Methods" for detailed modeling procedures). Such a model can generate oscillatory gene expression dynamics, consistent with previous theoretical and experimental studies of the circuit (Elowitz and Leibler, 2000; Loinger and Biham, 2007). When simulating the circuit from two different initial conditions, we generated time trajectories that gradually converge to a limit cycle in different ways (Fig. EV1B,C). For each case, we applied STICCC to the gene expression snapshots extracted from the simulated time trajectories, and we found the inferred outgoing transition vectors form a clockwise circular flow in the PCA projection, largely consistent with the directions along the simulation trajectory (Fig. EV1B,C). Interestingly, the linear-regression-based algorithm successfully recovered the state transition patterns even when the circuit gene expression dynamics were analyzed with nonlinear differential equations.

Next, we simulated the same circuit but using RACIPE, an ODE-based systems biology modeling framework for generating an ensemble of models with randomly sampled parameter sets from a GRN topology. The simulated gene expression data from RACIPE can be analogized to single-cell gene expression measurements with large cell-to-cell variability (see "Methods") (Katebi et al, 2020; Huang et al, 2017; Kohar and Lu, 2018). Unlike the above-mentioned simulations that exhibit oscillatory gene expression dynamics, most of the simulated expression profiles from RACIPE correspond to the stable steady states of dynamical models. Even for such a noisy dataset, when supplied as the input to STICCC, they yield a similar circular pattern of outgoing transition vectors (Fig. EV1D), consistent with how REP operates as an oscillatory circuit. The observed flow was also robust to variations in dataset size: even in down-sampled sets of as few as 200 simulated models, the overall predicted vector field remains the same (Fig. EV1E). In addition, transition vectors for this circuit were found to be dominated by tangential flow only beyond a certain radius from the origin (Fig. EV1F). These results together demonstrate the capability of STICCC in inferring dynamical state transitions from snapshots of gene expression profiles.

### STICCC captures the net flow and basins of attraction of synthetic circuits

Whereas $v_1$ represents the likely future expression changes of cell $i$, STICCC also infers its likely prior states by an incoming transition vector $v_2$, which evaluates how CCC values from neighboring cells to cell $i$ change with respect to gene expression differences (Eq. (9) in "Methods"). Since the vectors represent the outgoing and incoming transitions of the same cell, the average of $v_1$ and $v_2$ reflects the *net flow* resulting from irreversible state transitions, while $(v_1 - v_2)/2$ indicates the *reversibility* of state transitions (Fig. 1F). This approach can therefore be applied to a range of systems and describe both reversible and irreversible cell state transitions.

We applied STICCC to evaluate the inference of reversible and irreversible cell state transitions on several small synthetic gene regulatory circuits, where circuits' dynamical behaviors have been well-studied and intuitive. To compare the vector fields to observed gene expression distributions, we also identified basins of attraction using a heuristic approach based on the probability density (see

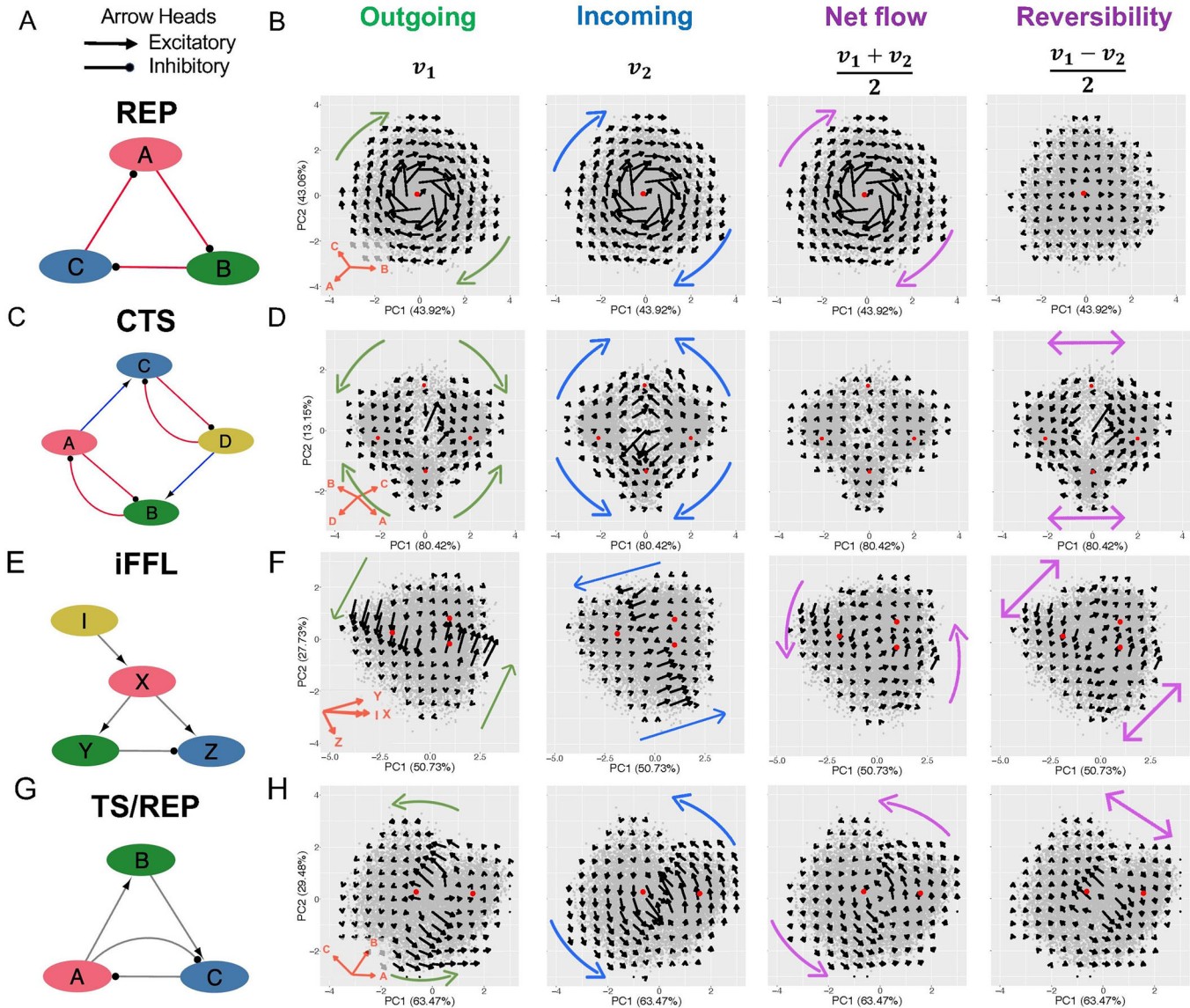

**Figure 2. Reversible and irreversible state transitions of synthetic circuits.**

Application of STICCC to synthetic circuits. Each row shows the outgoing vectors (2nd column), incoming vectors (3rd column), net flow (4th column), and reversibility (5th column) for each synthetic circuit (diagram in 1st column). The four synthetic circuits are repressilator (REP, **A**, **B**), coupled toggle switch (CTS, **C**, **D**), incoherent feedforward loop (iFFL, **E**, **F**), and a toggle switch/repressilator (TS/REP, **G**, **H**). For each, 10,000 simulated gene expression profiles are projected onto the first two principal components (PCA loadings are shown in the 2nd column). Colored arrows illustrated the overall transition patterns for the outgoing vectors (green), incoming vectors (blue), and net flow/reversibility (purple), based on visual inspection. Red dots in (**B**, **D**, **F**, **H**) indicate the estimated position of gene expression basins based on density.

"Methods", Appendix Fig. S1A). The first example is the same REP circuit we have discussed (Fig. 2A). We simulated single-cell gene expression data for a population of cells by RACIPE (Huang et al, 2017) (see "Methods"). Using the ensemble of simulated steady-state gene expression values, STICCC predicts a continuous clockwise oscillation for both the outgoing vectors $v_1$ and the incoming vectors $v_2$ (Fig. 2B). Since the directions and magnitudes for the $v_1$ and $v_2$ vectors are mostly the same, the net flow is very similar to either vector. Meanwhile, the reversibility vectors have almost zero magnitude, supporting the expected directed flow observed in the REP circuit. A noteworthy feature of STICCC is its

ability to robustly capture clear patterns of the irreversible state transitions despite noise and variability represented in the simulated models.

Whereas STICCC reveals oscillatory patterns and irreversible transitions in the REP circuit, it can also characterize multi-stable circuits with reversible transitions. We next simulated single-cell gene expression for a coupled toggle switch (CTS) (Fig. 2C; Appendix Fig. S1B). The circuit permits two major gene expression states, with high expression of one pair of genes, A/C, or another, B/D, in addition to two minor gene expression states characterized by high expression of A/D and B/C. Interestingly, the $v_2$ vectors for

the CTS circuit showed the opposite direction of transition as $v_1$, making the magnitude of net flow near zero (Fig. 2D). The reversibility is highest in the regions of state transitions to and from the two major expression states, suggesting the circuit allows bidirectional transitions between the two major gene expression states (high expression of A/C and B/D, respectively). Cells in the two minor states (high expression of A/D and B/C, respectively) were predicted to move toward one of the two major states, and the magnitude of the $v_1$ vectors diminishes as models move from the center of the PCA plot towards the center of each basin. The density-based analysis of gene expression basins confirmed the presence of these states, with peaks corresponding to the predicted transition pattern from STICCC.

Next, we applied STICCC to an incoherent feedforward loop (iFFL), a gene circuit known to generate gene expression dynamics allowing excitation and adaptation (Mangan and Alon, 2003) (Fig. 2E; Appendix Fig. S1C). In this iFFL, gene X directly activates but indirectly inhibits downstream target Z through gene Y; the circuit allows a pulse-like response, where Z expression initially increases in response to X, then decreases as the indirect regulation from Y takes effect (Mangan and Alon, 2003). These time-dynamic behaviors are not apparent from simulated steady-state gene expression of a cell population alone, with no observable temporal patterns. Using STICCC, the $v_1$ and $v_2$ vectors capture state transitions in different regions of the phenotypic space, such that the net flow creates an oscillatory transition, while the reversibility vectors show two-way transitions between two basins (Fig. 2F). This combination of irreversible flows and basins of attraction explains the typical dynamics of an iFFL as follows. When the signaling node I increases, the cells would transit from the bottom-left basin to the top-right basin through the counterclockwise oscillatory path (Appendix Fig. S2 and Appendix Table S1). This transition path allows Z expression to return to its original level from a deviation caused by the initial signaling in I. Interestingly, analysis of the density distribution revealed three distinct basins, but consistently high density in the space between the two basins vertically aligned on the right side suggests the first two principal components are insufficient to capture the true structure of the data (Appendix Fig. S1). The pulse-like behavior of an iFFL has been previously characterized using traditional systems biology modeling (Mangan and Alon, 2003), and here STICCC can capture such dynamical behaviors well with single-cell transition vectors.

The last synthetic circuit comprises a combination of a toggle switch motif (genes A and C) and a third gene B, allowing a negative feedback loop (Fig. 2G; Appendix Fig. S1D). Thus, this circuit, termed the toggle switch/repressilator (TS/REP), couples a bistable motif and an oscillatory circuit motif. However, the simulated single-cell gene expression distribution showed only two slightly separated basins in the gene expression space (Fig. 2H). On the other hand, the STICCC-inferred vectors exhibit an oscillation in the net flow, as well as reversible transitions between two basins. The net flow illustrates that the transitions between two basins follow two distinct paths according to the basin in which they begin (indicated by the purple arrows in the third panel from the left in Fig. 2H).

In conclusion, from the applications to these classic synthetic gene circuits, STICCC can reveal their dynamical behaviors, such as properties of basins of attraction, transition paths between states, and oscillatory dynamics, most of which are not apparent from snapshots of single-cell gene expression. Note that STICCC requires the regulator–target relationship of the gene regulatory circuit, but not the detailed rate equations and kinetic parameters.

## STICCC is robust against technical noise in scRNA-seq data

The simulated gene expression steady states of a GRN were generated by RACIPE, which captures the effects of cell-to-cell variations by using a large range of kinetic parameters. The simulations did not explicitly model the technical noise introduced by sampling transcripts in scRNA-seq data, however. As such, the RACIPE simulation data for the REP and CTS circuits were post-processed to model dropouts using methods from the single-cell transcriptional regulation simulator BoolODE (Pratapa et al, 2020). To evaluate the overall change in vector fields caused by simulated noise, we varied the dropout proportion and measured the pairwise change in angle for the predicted vector for each cell (net flow for the REP, reversibility for the CTS) from the original simulation, as well as a null distribution constructed from shuffling the indices of the original simulation. At low dropout proportions (<30%), the angle changes remain relatively small for both the REP and CTS circuits. However, the CTS showed a larger deviation in transition vectors due to noise than the REP at higher dropout proportions (Fig. 3A,G). In both circuits, the effects of noise grow in proportion to the dropout rate, but the qualitative results are well preserved, even at higher noise levels (Appendix Fig. S4). Our results suggest that the reliance of STICCC on not one cell, but a local distribution of samples, helps to mitigate the effects of technical noise in single-cell data.

## STICCC predicts state transitions observed in stochastic dynamical trajectories

STICCC predictions were also validated against time trajectories from the REP and CTS models. For REP, kinetic parameters of the differential equations were selected to produce a limit cycle, and the time trajectory from random initial conditions was simulated with a low noise level for a long duration (achieved by modeling the corresponding stochastic differential equations in Eq. (2) in "Methods"; see model parameters in Appendix Table S2). A random subset of 2000 snapshots from this noisy trajectory were used as the input expression data for STICCC. We then sampled 76 points along the deterministic limit cycle and compared the predicted vector from the local neighborhood of each point to the distribution of observed vectors (Fig. 3B, arrows indicate predicted net flow). To calculate observed vectors, we selected points from the noisy time trajectory that were close to a particular point along the limit cycle, then found the corresponding gene expression states after a short time lag (see "Methods"). An observed vector was defined as the vector from the state at one initial timepoint to the state after a short time delay, i.e., a few steps forward in the trajectory. The STICCC predictions closely agree with the observed vectors and the general direction of the limit cycle (Fig. 3C, angles for net flow in purple, limit cycle in blue, and observed vectors on violin plots). Although there is variability due to noise and the slightly deformed shape of the limit cycle in the PCA projection, there is a consistent peak in observed vectors corresponding to the direction of the limit cycle. STICCC appears to slightly overestimate the angle, predicting an outward spiral instead of oscillation, which may reflect the influence of noise in creating larger oscillatory trajectories than observed in deterministic simulations.

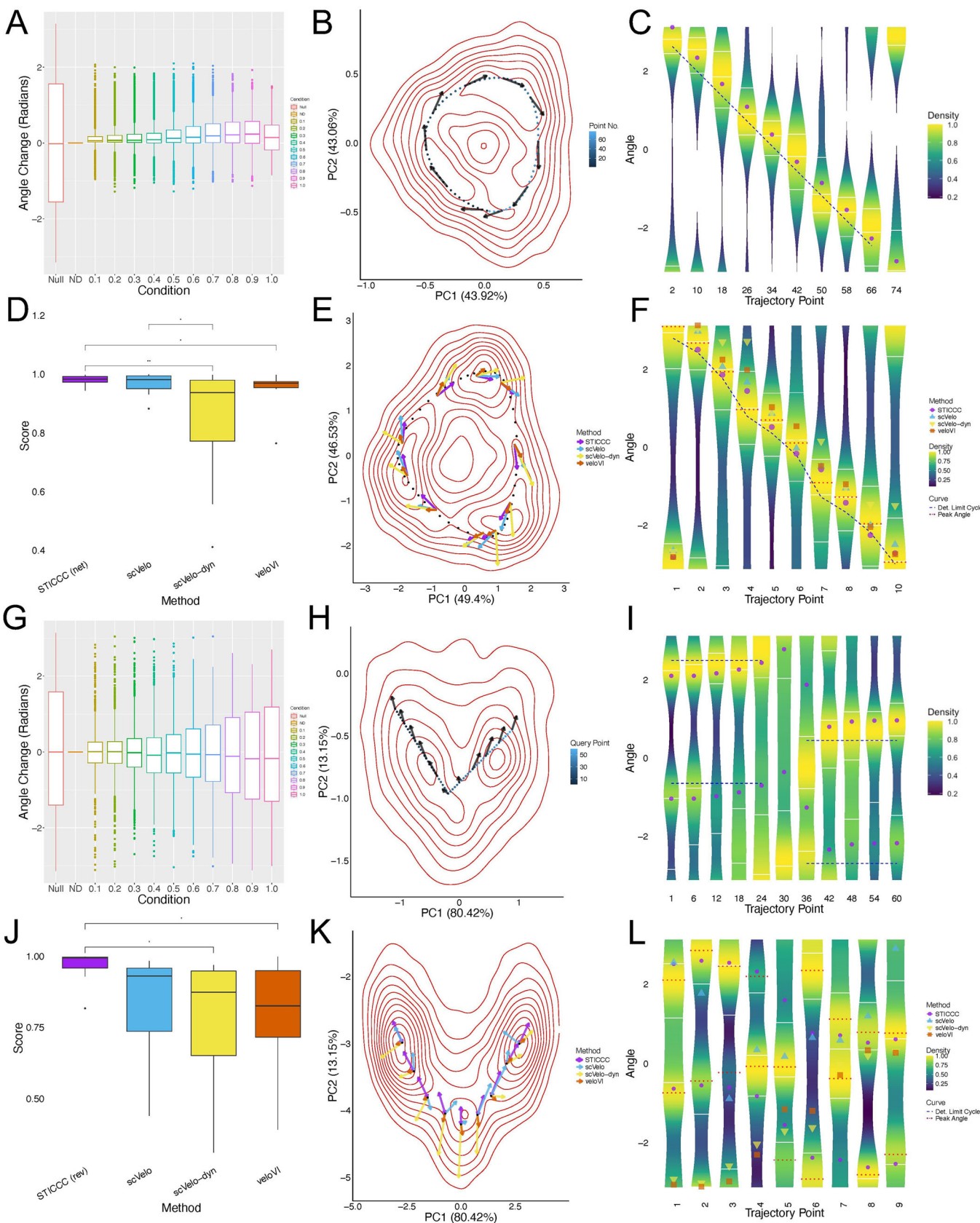

◄  **Figure 3.  State transition inference is robust against dropout and agrees with simulated expression trajectories.**

Single-cell gene expression data with various levels of dropout effects are simulated and applied to STICCC, then compared with RNA velocity methods for the repressilator (REP, **A–F**) and coupled toggle switch circuits (CTS, **G–L**). (**A**) Boxplots showing the distribution of angle changes of the single-cell vector predictions for various dropout rates in a simulation of REP models ($n = 1000$ per noise level), indicated on the $x$ axis and by color, compared to those for the condition with no dropout (labeled as ND). Null denotes the case when changes are computed by comparing the vectors from the ND condition between random pairs of cells. Other cases denote the conditions for different dropout levels α, where dropout simulations were performed using BoolODE, which sets read counts below a specified threshold (*dropQuantile*) for each gene to zero with a probability of α as shown in the $x$ axis and colors, and *dropProb* was fixed at 0.2. The parameter *dropProb* was specified as α shown in the $x$ axis and colors, and *dropQuantile* was fixed at 0.2. Boxes encompass the interquartile range (IQR) and mark the median with a horizontal line, and whiskers extend to 1.5 times the IQR. (**B**) PCA density plot for the stochastic trajectory of a REP model with a limit cycle. Contours (in red) indicate density along the whole trajectory ($n = 2000$); points indicate the deterministic limit cycle and are numbered 1–76 (indicated by light to dark blues). Arrows at several points illustrate the net flow predicted by STICCC using the simulated gene expression and the REP network. (**C**) Violin plot showing the distribution of observed angles (in radians) starting from each labeled point in the deterministic limit cycle ($n = 100$ per column). $X$ axis shows start points, violins show the distribution of angles in the first two PCs of observed transition vectors, the blue dashed line indicates the tangential directions of the limit cycle trajectory, and the purple dots are the angles of the predicted net flow. (**D**) Boxplot of cosine similarity scores between prediction methods over sampled points from REP stochastic simulations ($n = 50$), with method denoted by fill color. Statistically significant comparisons identified with the paired Wilcoxon test are annotated with asterisks (*$P < 0.05$, **$P < 0.01$). (**E**) PCA density plot for stochastic trajectory of a REP model with RNA splicing ($n = 5000$). Contours indicate density as in (**B**), points indicate the deterministic limit cycle, and vectors are shown from scVelo (blue) and STICCC net flow (purple). (**F**) Violin plot showing observed angles from splicing simulation, as in (**C**) ($n = 100$ per column). Predicted angles are denoted by overlaid points, with the method identified by color and shape. The blue dashed line indicates the angle between sampled limit cycle points, and red dashed lines indicate the most likely observed angles at each point. (**G**) Boxplots showing angle changes for various dropout levels for simulated CTS models, as in (**A**) ($n = 1000$ per noise level). (**H**) PCA density plot for stochastic trajectory of a CTS model switching between states ($n = 2000$). Reference points were selected along a linear path connecting basins, as numbered by 1–60 (by light to dark blues). Arrows are shown at selected points along the path to indicate reversibility predictions by STICCC. (**I**) Violin plot of the observed and predicted angles along the path ($n = 100$ per column). For each starting point indicated on the $x$ axis, the violins show the distribution of observed angles from trajectory simulations, the blue dashed lines indicate the forward and backward directions along the path, and the purple dots indicate either end of the predicted reversibility vectors. (**J**) Boxplot of cosine similarity scores between prediction methods over sampled points from CTS stochastic simulations ($n = 10$). Statistically significant comparisons identified with the paired Wilcoxon test are annotated with asterisks (*$P < 0.05$, **$P < 0.01$). (**K**) PCA density plot for stochastic trajectory of a CTS model with RNA splicing ($n = 5000$). Contours indicate density as in (**G**), points are selected along a path between basins, and arrows show scVelo (blue) and STICCC reversibility (purple) vectors. (**L**) Violin plot of observed and predicted angles for CTS splicing simulation, as in (**H**) ($n = 100$ per column). Red dashed lines indicate the most likely observed angles for each sampled point.

To validate the CTS predictions, we simulated time trajectories for a specific CTS model with three steady states and sufficient noise to drive stochastic state transitions between basins (Appendix Table S3 for model parameters). Similarly, a random subset of 5000 snapshots from this noisy trajectory was used as the input expression data for STICCC. We chose a set of reference points along a linear path between the three basins to compare observed and predicted vectors (Fig. 3H, arrows indicate predicted reversibility). Observed vectors generally showed a bimodal distribution, with trajectories continuing into either basin. These observed peaks aligned with either end of the predicted reversibility vector, suggesting STICCC can capture bidirectional transition routes (Fig. 3I, vector angles for linear path in blue, bidirectional reversibility in purple, and distribution of observed vector angles on violin plots). In summary, the reversible and irreversible state transitions predicted by STICCC are largely consistent with the local state transitions observed in stochastic simulations of the circuit models.

We applied a similar scheme using simulated data to compare the directions predicted by RNA velocity (calculated using the package scVelo (Bergen et al, 2020), using both its default mode and dynamical mode, as well as the more recently published tool veloVI (Gayoso et al, 2024)) to those from STICCC in the REP and CTS circuits. In the REP case, RNA velocity vectors and STICCC net flow vectors both agreed with the direction of the deterministic limit cycle and were well aligned with observed trajectories in stochastic simulations (Fig. 3E,F). On the other hand, for the CTS circuit, whereas RNA velocity vectors pointed toward each of three states visited by a stochastic trajectory, STICCC showed very little net flow, with larger reversibility vectors aligned with the transition paths between states (Fig. 3K,L). The similarity between observed and predicted vectors varied along the reference trajectories, but STICCC achieved the highest similarity in general (Fig. 3D,J; Appendix Fig. S3A,B) (see "Methods"). We also compared the

results from STICCC to RNA velocity vectors over all points in each dataset, finding that the similarity across all three RNA velocity methods was relatively high, and they generally agreed with predictions for STICCC's net flow in the REP circuit and reversibility in the CTS circuit (Appendix Fig. S3C,D). These results on simulated data suggest that while RNA velocity and STICCC are both able to identify transition directions for single cells, STICCC uses information from the GRN, rather than from unspliced RNA, to separate irreversible transitions from reversible transitions in multi-stable systems.

## Inferring signal-induced state transitions

We next evaluated whether STICCC would predict single-cell state transitions of a system with a varying signal input. We again simulated the CTS circuit (Fig. 4A), but this time beginning from the cells from the leftmost basin (Fig. 4C, $t = 0$), and performed time dynamic simulations for each model with a varying signaling state (Fig. 4B). An increased signal (increasing when $t \in [0, 20]$ and constant when $t \in [20, 40]$) would cause up to a 50-fold increase of the production rate of gene D, which can drive the cells to transit to the rightmost basin. In the simulation, we added a small amount of stochastic noise to permit cells to escape from the initial basin (see "Methods"). A decreasing signal (when $t \in [40, 60]$), on the other hand, should negate the effects, allowing some models to transit back to the original state. Gene expression snapshots were extracted at selected timepoints ($t = 0, 1, 3, 40, 60, 80$) and supplied as inputs to STICCC with the same circuit topology. In the presence of an increased signal, as the cells transit to the target state, the vector fields gradually shift as all basins of attraction are drawn in the direction of the perturbed gene (Appendix Fig. S5). A small net flow towards the target state appears during signaling (Fig. 4C, $t = 3, 40, 60$). Meanwhile, reversibility vectors shifted under

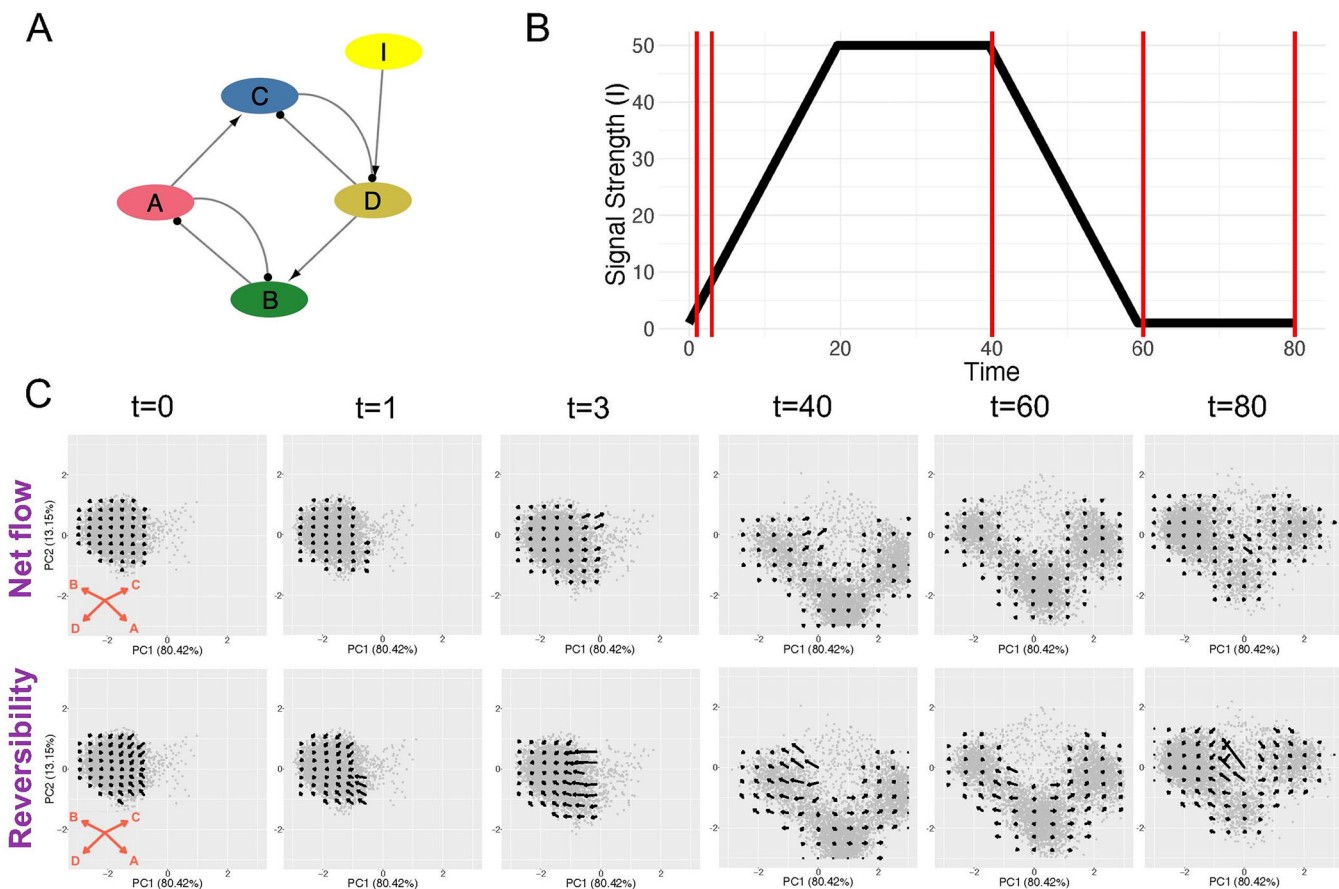

**Figure 4. Characterization of signal-induced state transitions from snapshot data.**

(A) The CTS circuit was modified to include a manually controlled input signal I, which regulates the production rate of gene D. (B) Time-dependent signal strength. Production rate of gene D increased linearly up to 50-fold during t ∈ [0, 20], then remained at the high level during t ∈ [20, 40] before linearly decreasing back to baseline levels during t ∈ [40, 60]. (C) Net flow and reversibility at selected timepoints, indicated by vertical red lines in (B). Data is projected to the same PCA axes as the original CTS results. A small net flow toward the target state appears during signal induction, and the apparent separation between basins suggested by the reversibility vectors shifts to the left, both of which changes largely revert upon signal removal.

signaling such that the boundary between apparent basins moved further to the left (Fig. 4C, compare $t = 40$, 60). This corresponds to the expected effect of driving cells toward the B/D state, as a shorter duration and smaller amount of noise will be sufficient to cause transitions in the favored direction. Moreover, these changes began to revert as the signal was decreased and eventually removed, such that the portion of the vector field shown at $t = 0$ is comparable to the same region at $t = 80$. As shown in this more challenging test where the signaling information is not included in the input of the algorithm, STICCC can clearly detect the signal-induced state transitions from snapshots of single-cell gene expression data and circuit topology.

## Inferred vectors characterize state transitions from scRNA-seq data

Having tested the method on simulated datasets, we applied STICCC on three single-cell gene expression datasets. First, we applied STICCC to an scRNA-seq dataset of 976 budding yeast cells. For the input regulatory links, we used an established GRN

model of the yeast cell cycle (Fig. 5A) (Katebi et al, 2020; Data ref: Jackson et al, 2019). Dimensional reduction of 5624 genes measured in the scRNA-seq data using PCA revealed a circular structure that arranged cells in order of cell cycle phase annotations obtained from Seurat (see "Methods") (Fig. 5B). Although the topology includes various types of regulation, including transcriptional and signaling relationships, in this case, we considered each interaction equally when computing cross-cell correlations. Since some of the genes in the GRN do not correspond to a single transcript, we applied the same mapping as in Katebi et al, 2020 to select genes to retrieve gene expression data (Appendix Table S4). After applying STICCC, we found a circular pattern where the net flow forms a path along the order of cell cycle phases, with little reversibility overall.

To uncover the potential roles of genes and interactions in different types of cell state transitions, one can also supply a modified topology alongside the same single-cell gene expression data and compare the resulting vectors to the original results (see "Methods"). Applying this approach to synthetic circuits suggested that minor changes to the input GRN tend not to drastically change

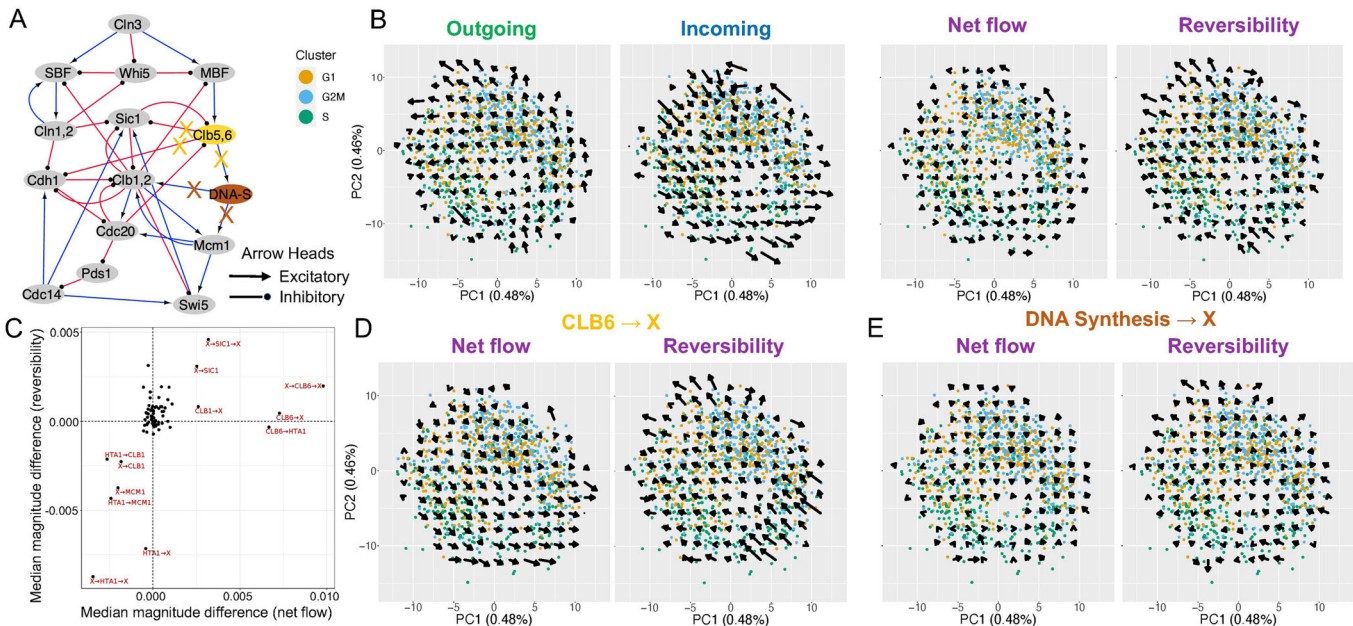

**Figure 5. Velocity inference in an experimental scRNA-seq data of budding yeast cell cycle.**

(A) The gene regulatory network topology for the budding yeast cell cycle, with blue pointed arrowheads denoting activation and red circular arrowheads denoting inhibition. (B) The inferred outgoing, incoming, net flow, and reversibility vectors. Each plot shows the projection of gene expression profiles of network genes onto its first two principal components. Colors of points represent the cell cycle states. (C) Summary of edge-sensitivity analysis showing median magnitude difference between paired vectors across edge perturbations, where vectors are computed with part of the GRN intentionally left out. X axis shows the differences in net flow, and y axis shows the differences in reversibility. Red labels highlight the perturbations with significant changes in either net flow or reversibility. (D) Net flow and reversibility with outgoing edges from CLB6 omitted. (E) Net flow and reversibility with outgoing edges from HTA1 (DNA Synthesis) removed.

the output, although in some cases the effects are confined to a particular region of gene expression space in which the modified interactions are relevant (Appendix Figs. S6 and 7). Applying it to the cell cycle GRN, this edge-sensitivity analysis highlighted links relating to CLB6 and DNA synthesis as key components in producing the observed oscillatory transitions (Fig. 5C). CLB6 is known to regulate the initiation of S phase and removing related edges resulted in large changes to the predicted net flow, particularly near the cells in the S phase (Fig. 5D). Whereas edges linked to DNA synthesis were removed, reversibility overall was greatly impacted, suggesting the importance of these edges in the dynamical behaviors of the cell cycle GRN (Fig. 5E). Prior work has suggested these genes and edges are significant actors in the control of the yeast cell cycle (Adler et al, 2022), suggesting the power of the edge-sensitivity analysis in uncovering important network components, especially those that are nonredundant in the network. Notably, this approach permits uncovering influential nodes and edges from a topology without any detailed physical information about the nodes, implying that STICCC could effectively identify key regulators in less-known systems. Moreover, the relatively small impact of most network modifications implies that STICCC can provide information about a GRN's behavior even when the GRN is not fully comprehensive. In summary, STICCC effectively captures the irreversibility and oscillatory nature of cell cycle stages, as well as indicating which regulatory information most informs the predictions.

We cannot apply RNA velocity to the above dataset because of a lack of RNA splicing activity in the yeast genome. To directly

compare STICCC with RNA velocity, we also examined a similar dataset describing the cell cycle in mammalian cells, using a modified network topology with human genes (see "Methods" and Fig. EV2A). Here, we obtained a dataset of gene expression in U2OS cells, which has been studied before with a number of techniques, including RNA velocity, which yielded a circular flow of vectors aligned with cell cycle phases (Mahdessian et al, 2021; Zhu and Wang, 2024; Data ref: Mahdessian et al, 2020). Similar to the yeast cell cycle result, we observe net flow vectors pointing from G1 to S, S to G2M, and G2M back to G1 (Fig. EV2B,C), indicating a similar structure of attractors and oscillatory behavior is present in the mammalian cell cycle GRN as in yeast. Interestingly, some vectors appear to point backward relative to the expected progression of cell cycle phases, particularly in the region between G1 and G2M, suggesting some cells may follow a different trajectory and remain in a G1-like state rather than progressing through the cell cycle. To assess the relative speed of predicted transitions, we compared the magnitude of net flow and reversibility vectors, as well as the relative proportion of cell cycle phases, according to the angle of cells relative to the origin. The ratio of net flow to reversibility reached minima near the points where the dominant phase changes, generally followed by a sharp increase in the same ratio. This pattern may indicate regulatory checkpoints that slow net flow and must be overcome to allow irreversible cell cycle progression, which then unfolds more quickly (Fig. EV3).

The next example is a single-cell qPCR dataset (Mojtahedi et al, 2016; Data ref: Mojtahedi et al, 2016) describing hematopoietic

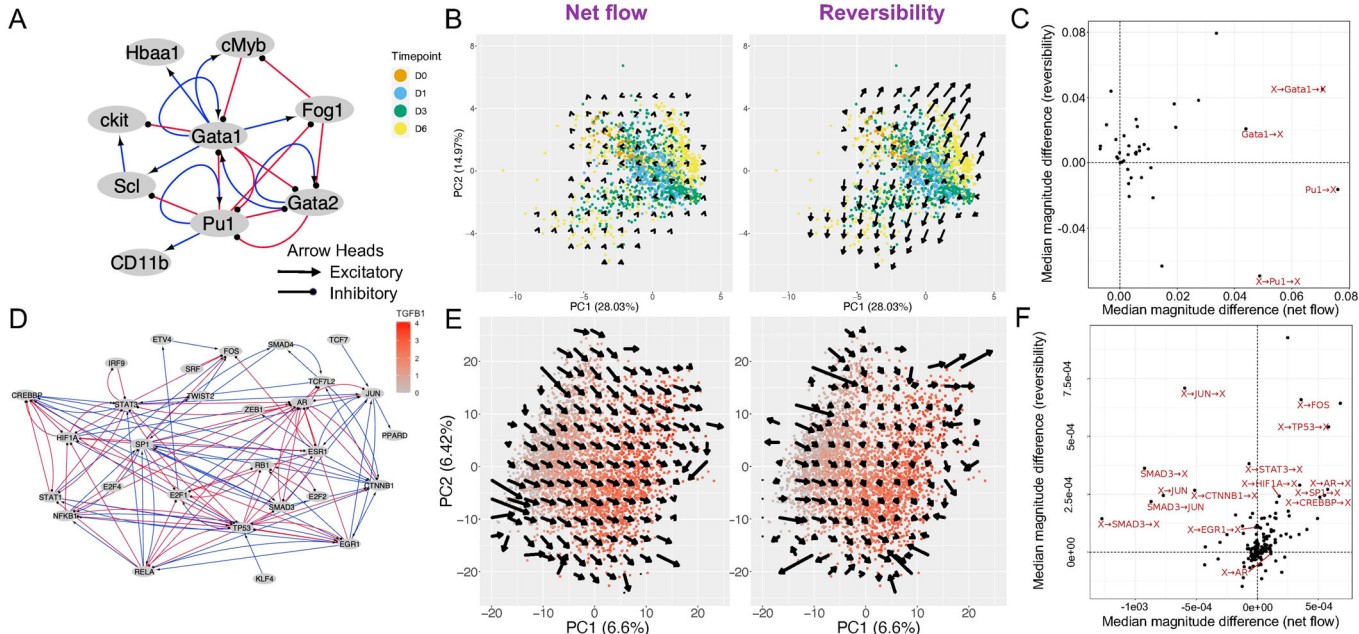

**Figure 6. STICCC distinguishes cell state transitions with dominant net flow or reversibility.**

(A) Gene regulatory network topology for differentiation of hematopoietic stem cells. (B) PCA of HSC gene expression data colored by timepoint, with grid-smoothed vectors for net flow (left panel) and reversibility (right panel). Fig. EV2 shows similar results for each treatment condition analyzed separately. (C) Summary of influential edges and nodes from edge-sensitivity analysis. Edge perturbations are labeled in red if they are in the top 15% of perturbations by combined change in median net flow and reversibility. (D) NetAct inferred transcription factor regulatory network using the scRNA-seq data from A549 cells treated with TGF-β undergoing epithelial-mesenchymal transition. (E) PCA plots of A549 gene expression data during EMT along with smoothed arrows indicating net flow (left panel) and reversibility (right panel) predictions. (F) Summary of influential edges and nodes for EMT network, with the top 15% of perturbations labeled in red. X axis denotes the median paired change in vector magnitude for net flow, and y axis denotes the same quantity for reversibility.

stem cells (HSCs) differentiated into either erythroids or myeloids using erythropoietin (EPO), granulocyte macrophage-colony stimulating factor (GM-CSF), and a combination of both drugs simultaneously. Gene expression was measured over 6 days after signal induction, and the stem cells bifurcated into two differentiated states. A GRN for HSC differentiation has been previously established through literature curation (Mojtahedi et al, 2016) and served as the basis for which genes to measure in the qPCR. In our analysis, from the GRN genes, we further removed genes with low expression and variance (see "Methods", Fig. 6A). PCA on the combined dataset comprising cells receiving all three treatments revealed a bifurcating structure, where early timepoints for all treatments clustered together with untreated cells, then the populations diverged into the two differentiated cell lineages based on the treatment at later timepoints (Fig. 6B). The resulting net flow from STICCC had very small magnitudes, but the reversibility vectors indicated a bifurcation occurring around data from day 3 into two differentiated states around data from day 6. This is consistent with findings from the original work that the acquisition of erythroid- and myeloid-specific markers occurred mainly between day 3 and day 6 (Mojtahedi et al, 2016). Strong reversibility vectors suggest the existence of multiple basins of attraction that are associated with various cell phenotypes during cell differentiation. The reversible cell state transitions may also be related to the rebellious cell states, where some cells, upon symmetric destabilization of a progenitor, differentiate into the opposite phenotype of the inducing signal. Interestingly, we did not

observe single-cell vectors associated with the state transitions during the early timepoints (before day 3) (Mojtahedi et al, 2016), suggesting that the network may miss gene regulatory interactions involved in destabilizing the progenitor state. Moreover, similar patterns of vector fields were observed when STICCC was applied to data for each treatment condition separately (Fig. EV4). Finally, edge-sensitivity analysis of the circuit revealed Gata1 and Pu1 as highly influential (Fig. 6C), which is supported by experimental evidence indicating these genes as critical supporters of the erythroid and myeloid lineages, respectively (Burda et al, 2010).

Lastly, we applied STICCC to scRNA-seq data characterizing TGF-β-induced epithelial-mesenchymal transition (EMT) in a published dataset of 3133 A549 cancer cells, sequenced at five timepoints during the course of 7 days' exposure to TGF-β (Cook and Vanderhyden, 2020; Data ref: Cook and Vanderhyden, 2020). The GRN was obtained from the method NetAct, which integrates information from transcription factor (TF)-target gene databases and TF activity inference to construct a dynamic transcriptional regulatory network (Fig. 6D) (Su et al, 2022). STICCC predicts a much stronger net flow than reversibility, where the net flow vectors consistently moved from the E population to the M population, as would be expected from a strong EMT-inducing signal (Fig. 6E, cells colored by TGFBI expression). TGF-β-induced EMT has been characterized as irreversible (Tian et al, 2013), although recent work suggests reversibility may occur on longer timescales through epigenetic mechanisms not included in the present network (Jain et al, 2023). Edge-sensitivity analysis

highlighted AP-1 TF family members Jun (Piechaczyk and Farràs, 2008) and Fos (Kovary and Bravo, 1991), as well as Smad3, a well-known mediator of the TGF-β signaling employed in the dataset (Meng et al, 2018) (Fig. 6F). Overall, STICCC appears to perform well on experimental single-cell data, capturing known state transitions in the context of multi-step differentiation, bifurcation, and oscillatory dynamics. We expect STICCC to allow to generate new hypotheses regarding cell state transitions and individual genes or edges driving them.

## Discussion

In this study, we have presented a computational method for inferring reversible and irreversible cell state transitions from single-cell gene expression data and a set of regulatory interactions, which provides insight into the gene regulation of phenotypic state transitions. We first developed the method in the context of small synthetic circuits and simulated single-cell gene expression data, where the circuits' dynamics are known. We found that the structure of the governing GRN encodes sufficient information to recover oscillatory trajectories and multi-state distributions. Moreover, the incoming and outgoing cross-cell correlations (CCCs) provide complementary information, helping to distinguish reversible and irreversible interactions for a more complete characterization of cell state transitions. STICCC can accurately recapitulate the expected behaviors of multiple synthetic circuits, produce robust results in noisy conditions, and uncover the roles of different nodes and edges by edge-sensitivity analysis. Moreover, STICCC can detect reversible and irreversible state transitions in a system driven by an external signal, even when the signal interaction is not presented in the input circuit topology. For simulated gene circuits, STICCC reliably uncovers reversible and irreversible transition patterns and can describe the structure of phenotypes in multi-stable, oscillatory, or hybrid systems.

Following the success of STICCC in characterizing simulated data, we applied it to three experimental single-cell gene expression datasets. For the cell cycle dataset, the method recovered the expected direction of oscillation and identified varying levels of irreversibility among cell cycle phase transitions. Edge-sensitivity analysis also uncovered significant drivers of cell cycle progression, which allows to generate new hypotheses and suggest targets for genetic perturbation. STICCC also identified a bifurcation occurring during differentiation of hematopoietic stem cells and suggested an irreversible epithelial-mesenchymal transition during TGF-β induction in lung cancer cells. STICCC shows great promise to explore and decode gene networks governing complex cellular processes.

Some existing methods address similar questions, notably including RNA velocity and related work, which is a set of methods aiming to predict short-term changes in gene expression by analyzing the presence of spliced and unspliced mRNA transcripts (La Manno et al, 2018; Qiao and Huang, 2021; Bergen et al, 2020; Qiu et al, 2022; Cui et al, 2024). Our approach differs from RNA velocity in several key aspects as follows. First, STICCC incorporates prior knowledge of the GRN, rather than a kinetic model of mRNA splicing, to generate transition vectors. Second, STICCC directly relies on a local distribution of samples to compute each vector, mitigating some effect of noise in the input

data. Direct comparisons on simulated trajectories show that STICCC and RNA velocity capture similar features of GRN behavior, but STICCC critically separates reversible and irreversible transitions, giving insight into state transitions in multi-stable systems. This framework enables STICCC to describe transitions where cells in a dataset may be in transit in either direction, where RNA velocity may be confounded by contradictory vectors. Other recent work including scKINETICS (Burdziak et al, 2023) and TFvelo (Li et al, 2024) incorporate regulatory information in trajectory inference, using TF-target relationships as the basis for a dynamical model and constraining velocity predictions based on observed gene expression states and/or chromatin accessibility. Whereas these methods incorporate linear models of regulatory activity, however, we demonstrate that STICCC can predict state transitions from gene expression data associated with nonlinear regulatory dynamics. Moreover, we are not aware of existing work that explicitly addresses the question of disentangling reversible and irreversible state transitions, while STICCC provides a convenient framework for this.

We anticipate applications of STICCC in studying the dynamics and basic properties of GRN topologies. By systematically applying the algorithm to small circuit motifs, one could characterize circuits according to the prevalence of reversible or irreversible flows, expanding on other recent work addressing essential properties of circuit motifs (Huang et al, 2022; Ahnert and Fink, 2016; Ye et al, 2019; Shen-Orr et al, 2002; Jiménez et al, 2017). Systematic patterns in the relationship between GRN topology and predicted flows could help to predict state changes in experimental data based on inferred and a priori known interactions. Conversely, the same mapping between circuits and dynamical behaviors may help to determine the regulatory circuitry underlying observed transition data. Influential edges and nodes can be prospectively identified and validated with perturbation studies to efficiently identify key regulators and targets for intervention in disease systems (Yuan et al, 2021; Ben Guebila et al, 2022). STICCC is a step forward in analyzing cell state transitions in scRNA-seq data with the advantage of integrating measured data with known gene regulatory interactions.

The most important input to STICCC is the GRN topology, meaning its predictions are highly reliant upon having accurate knowledge of gene regulatory relationships governing the system of interest. While this inherently limits the scope of application to those systems where some prior knowledge is available, we note that STICCC appears to be able to describe the global features of a system even with minor inaccuracies in the input GRN (Appendix Fig. S6). Moreover, public databases of known regulatory interactions, as well as bioinformatics-based GRN reconstruction tools, may provide useful starting points where a fully validated GRN is not available. We show that STICCC can generate useful predictions with both inferred GRNs (as in the EMT case above) and literature-based GRNs (as in the cell cycle and HSC cases). Nonetheless, it should be noted that the quality of the input GRN can significantly influence the outputs of the algorithm. Moreover, the requirement of an input GRN assumes that a single GRN can describe any cell states present in the data; this may not hold for datasets with a population of many heterogeneous cell types. In addition to the input GRN, another possible source of error is the noise in gene expression data and the selection of the sampling radius parameter. While the method demonstrates some robustness

against these potential issues, these inputs may still introduce inaccuracy to the results.

There are several aspects of STICCC worth further investigation in the future. First, the computational challenge of making a stable inference while minimizing the time cost remains present. The current algorithm can analyze 10,000 models in a 4-node simulated circuit in ~10 min on a 2021 M1 MacBook Pro, but performance varies significantly with dataset size, distribution, and parameter settings. Re-implementation of major calculations in a more performant language such as C and parallelization of parts of the algorithm could yield significant improvements. Second, further work should be done to compare the reversibility predictions with high-resolution time-series scRNA seq data, which, while a rapidly growing resource (Chen et al, 2022), remains fairly sparse. Third, we also do not distinguish different types of regulations, e.g., signaling as opposed to transcriptional regulation, which may operate on different timescales and need to be considered separately in the analysis.

## Methods

### Reagents and tools table

| Reagent/ resource | Reference or source | Identifier or catalog number |
|---|---|---|
| **Software** | | |
| R-software v4.2.2 | R Core Team, 2022 | https://www.r-project.org/ |
| Python v3.9.0 | Python Software Foundation, 2025 | https://www.python.org/ |
| Cytoscape v3.10.3 | Shannon et al, 2003 | http://www.cytoscape.org |
| sRACIPE v1.7.3 | Kohar and Lu, 2018 | https://doi.org/10.18129/B9. bioc.sRACIPE |
| STICCC v1.0.0 | This study | https://doi.org/10.5281/ zenodo.18486460 |

### GRN simulations

To simulate scRNA-seq data, we applied random circuit perturbation (RACIPE (Huang et al, 2017), using sRACIPE (Kohar and Lu, 2018)) to generate a set of 10,000 simulated gene expression profiles for each GRN. RACIPE is an algorithm designed to generate, from a GRN topology, an ensemble of ordinary differential equation (ODE)-based models with randomly generated kinetic parameters. Each model represents a distinct set of parameters that could capture cell-to-cell variability, and collectively the simulation results yield a distribution of steady-state gene expression profiles similar to that of single-cell sequencing data (Kohar and Lu, 2018). In brief, the dynamics for a target gene $A$ regulated by genes $B_i$ are modeled in RACIPE as in Eq. (1) below:

$$\frac{dA}{dt} = \frac{G_A}{\prod_i \lambda_{B_iA}} \prod_i \left( \lambda_{B_iA} + \frac{(1 - \lambda_{B_iA})}{1 + (\frac{B_i}{B_iA_0})^{n_{B_iA}}} \right) - k_A A, \quad (1)$$

where $G_A$ and $k_A$ signify the maximum production rate and degradation rate of $A$, respectively, $\lambda_{B_iA}$ denotes the fold change of $A$ in response to regulation from $B_i$, $B_iA_0$ is the threshold level of regulator $B_i$, and $n_{B_iA}$ is the Hill coefficient. RACIPE assumes independent regulatory interactions, thus calculating the dynamics of a gene as the product of shifted Hill functions corresponding to each incoming regulatory interaction. Following the methodology of RACIPE, parameters are sampled from a uniform distribution, which ensures a high degree of variability in dynamics and steady states.

Noisy time trajectories were obtained via a similar approach, with stochastic differential equations with a Gaussian white noise term scaled to the average expression of each node, as described in (Kohar and Lu, 2018). The dynamics for a gene under such stochastic simulations are given by Eq. (2) below, where $\xi_A$ is the noise level for gene $A$ and $\eta(t)$ represents Gaussian white noise, with zero mean and unit variance:

$$\frac{dA}{dt} = \frac{G_A}{\prod_i \lambda_{B_iA}} \prod_i \left( \lambda_{B_iA} + \frac{(1 - \lambda_{B_iA})}{1 + (\frac{B_i}{B_iA_0})^{n_{B_iA}}} \right) - k_A A + \xi_A \eta(t). \quad (2)$$

Signal-induced state transitions (results in Fig. 4) were modeled by adjusting the parameter for production of the signaling target gene and performing new time-series simulations, using the steady-state solutions of the untreated condition as the starting point. In these perturbed simulations, the production rate parameter for the affected gene was multiplied by a factor $s(t)$, which linearly increases from the initial value of 1 to a specified fold change, then remains at the target signaling level before linearly decreasing back to the initial value (see Fig. 4B for the signaling dynamics). Simulations of signal-driven transitions took the form of Eq. (3):

$$\frac{dA}{dt} = s(t) \frac{G_A}{\prod_i \lambda_{B_iA}} \prod_i \left( \lambda_{B_iA} + \frac{(1 - \lambda_{B_iA})}{1 + (\frac{B_i}{B_iA_0})^{n_{B_iA}}} \right) - k_A A. \quad (3)$$

The package sRACIPE was also applied to simulate ODE time trajectories of specific individual simulated models. To do this, the method was applied as usual with the additional parameter printInterval, which recorded gene expression states at regular intervals as the ODE system evolved from random initial conditions. For the repressilator (REP) circuit (results in Fig. 3B,C, the parameters used to generate an oscillatory trajectory are listed in Appendix Table S2, and the parameters for the multi-stable CTS model (results in Fig. 3G,H are in Appendix Table S3. All simulation data from RACIPE were normalized using the included function sracipeNormalize, which simply log transforms simulated data with a pseudocount of 1 added.

### Dimensional reduction

We performed principal component analysis (PCA) on the log-normalized gene expression data to obtain a clear representation of the phenotypic state space. For datasets with a large number of genes sequenced, namely the EMT and cell cycle datasets used here, the first 15 principal components were used for vector inference instead of the gene expression matrix to save computational cost.

### STICCC algorithm

We develop a computational algorithm STICCC to infer the reversible and irreversible vectors of state transitions from single-

cell gene expression data and a set of gene regulatory interactions. First, the CCC between two cells $i$ and $j$, $\rho_{i \to j}$, is defined by a Pearson correlation (Fig. 1B)

$$\rho_{i \to j} = corr(e_{R,i} \odot s, e_{T,j}), \tag{4}$$

where $e_{R,i}$ and $e_{T,j}$ are the activity or expression levels of a set of regulators for cell $i$ and expression levels of a set of target genes for cell $j$, respectively; $s$ represents a sign vector indicating the interaction type (i.e., 1 for activation, and $-1$ for inhibition); $\odot$ denotes the Hadamard (or component-wise) product. The CCC represents the propensity of cell $i$ to tend towards the gene expression state of cell $j$. The transition propensity from cell $j$ to cell $i$, $\rho_{j \to i}$, can be then defined in a similar way but with the data source for regulators and targets reversed.

Next, from the single-cell gene expression profiles of a cell population, we infer a state transition vector for each cell by approximating as the local gradient of CCC with respect to the gene expression changes (Fig. 1C). Given a starting cell $i$ and a distribution of CCC values from cell $i$ toward its neighboring cells (i.e., cells within a set radius in gene expression space, see the section "Optimization of sampling radius" for the selection of a user-defined sampling radius), the outgoing transition vector $v_1$ of cell $i$ was computed using multiple linear regression

$$y_1 = v_1 X + \varepsilon, \tag{5}$$

where $X$ is an $n \times m$ matrix of relative gene expression values for $n$ neighboring cells from cell $i$. Each row of $X$, $X(j,:)$, is the gene expression difference or relative position between a neighboring cell $j$ and the starting cell $i$:

$$X(j,:) = x_j - x_i, \tag{6}$$

where $x_i$ represents either the normalized expression of $m$ network genes or the values of $m$ principal components or other user-supplied coordinates (e.g., t-SNE, UMAP) for cell $i$. $y_1$ is a column vector of size $n$ representing the relative incoming CCC values, where its $j$th component

$$y_{1,j} = \rho_{i \to j} - \rho_{i \to i}. \tag{7}$$

$\varepsilon$ is a vector for the noise term. Then, $v_1$ of cell $i$ was computed by Eq. (8) using the least-squares estimator

$$v_1 = (X^T X)^{-1} X^T y_1, \tag{8}$$

The outgoing transition vector represents the change in gene expression associated with the steepest increase in CCC, therefore suggesting a likely future expression state according to the GRN.

While $v_1$ illustrates the future state transitions of cell $i$, $y$ can also be replaced with $y_2$, a column vector of incoming relative CCC values from neighboring cells to cell $i$, as opposed to outgoing CCC from cell $i$ toward its neighbors ($y_{2,j} = \rho_{j \to i} - \rho_{i \to i}$). We define

$$v_2 = -(X^T X)^{-1} X^T y_2, \tag{9}$$

which indicates the direction of the transition from neighboring cells towards cell $i$. The minus sign in Eq. (9) is presented to allow to interpret $v_2$ as the direction of transition from cell $i$, similar to $v_1$ (Fig. 1C). In other words, whereas $v_1$ estimates the gradient of $\rho_{i \to j}$ to identify likely future states, $v_2$ estimates the reverse gradient of $\rho_{j \to i}$ to identify a transition from likely precursors. To integrate the information from outgoing and incoming transitions, we define the *net flow* as the average of $v_1$ and $v_2$, which captures irreversible transition patterns, and *reversibility* as $(v_1 - v_2)/2$, which captures bidirectional state transitions.

## Optimization of sampling radius

Since STICCC vectors are inferred based on the neighboring cells in gene expression space of each cell, it is important to carefully select the size of the neighborhood, which we represent as a fraction of the maximum pairwise distance in gene expression space between cells. A larger value will tend to include more cells in each neighborhood, helping mitigate the effect of noise in the gene expression data, but is more computationally expensive. Moreover, we found that the accuracy of the multiple regression deteriorated at larger sampling radii (Fig. EV5). To select an optimal sampling radius for a dataset, we aim to maximize the number of cells with a minimum neighborhood size of 15, a proportion we term 'coverage', while minimizing the error in the multiple regression. In particular, we calculated median absolute percentage error (MAPE) to ensure robustness to both outlying cells and variance in the magnitude of inferred vectors between datasets. In practice, we applied a simple grid search to test 11 evenly spaced sampling radii between 0.05 and 0.3, selecting that which maximizes the ratio of coverage to MAPE. Although a more sophisticated optimization may reduce regression error further, we found the transition patterns were relatively robust to small changes in the selected radius. Based on several simulated and experimental datasets, we observed an optimal search radius between 7-20%; in general, smaller, sparser datasets require a larger search radius to provide sufficient coverage. The default value of the search radius in STICCC is thus set to 0.15, or 15% of the maximum pairwise distance.

## Edge-sensitivity analysis

Given a starting GRN, the role of each member gene and interaction can be uncovered by removing them from the input topology and generating a new vector field on the same expression data, thus simulating a case where the ground-truth network is partially unknown or incorrect. The difference between the untreated and perturbed vector fields then shows the influence of the gene or interaction omitted. This can be systematically repeated to quantify and compare the roles of many genes, links, or submodules of the network. The overall impact of a change to the topology is summarized as the median pairwise difference in vector magnitude between the untreated and perturbed vector field – in this way, each perturbation can be compared on the axes of median pairwise magnitude change in net flow and reversibility.

## Comparison between STICCC predictions and simulated trajectories

To compare STICCC with simulated trajectories, we first generated noisy trajectories from SDE models with fixed parameters (see

Appendix Tables S2 and 3) and random initial conditions. Each circuit was simulated using sRACIPE for $10^5$ (for REP) or $10^6$ (for CTS) time units with a noise parameter of 0.1 for the REP and 2.0 for the CTS, sufficient to generate a robust gene expression distribution and many instances of the trajectory crossing near the same points. Given the full trajectory, we manually identified points in gene expression space to compare the directions of predicted and observed vectors. To identify observed directions, first all timepoints of the noisy trajectory passing within a set distance (2% of the maximum pairwise distance among the full trajectory data) of some query points were collected to serve as *start points*. For each start point, the subsequent gene expression state from the trajectory, after a specified lag period, was collected as an *end point*. The distribution of vectors from each paired start and end points was denoted the observed vector. The lag period was selected for each circuit to be long enough to overcome the effect of noise, but short enough to emphasize local, near-instantaneous gene expression changes, as well as to allow a similar variance in gene expression across instantaneous gene expression fluctuations versus slow overall state transitions. Specifically, this is achieved by computing the root mean squared distance (RMSD) in PCA space between start points and end points as a function of the lag time. Lag times were selected separately for each set of start and end points from the noisy trajectory such that the RMSD between start and end points was as close as possible to a target value, which in this study was set to approximately 0.65 (REP) and 3.5 (CTS) (Appendix Fig. S8).

To compare the predicted vectors with the observed trajectories, we compared the distribution of observed vectors near each query point as described above, with the vectors from STICCC in terms of cosine similarity. For a given reference point, we obtained the distribution of observed vectors and calculated the peak angles in the first two PCs using a circular von-Mises kernel density estimate, then identified the peak(s) of the distribution using R package *pracma* (Borchers, 2011). The cosine similarity score was calculated as the cosine similarity between a predicted vector and the nearest peak angle, in order to permit cases where heterogeneous trajectories are possible. These scores were then compared to a theoretical distribution of possible scores from uniformly sampling the range of possible angles.

## Comparison with RNA velocity on simulated trajectories

To generate an in-silico dataset for direct comparison with RNA velocity, we modified the simulations above for REP and CTS to include linear splicing of mRNA with a constant splicing rate of 0.7. The equation for these modified simulations is given by Eq. (10), which describes the dynamics of unspliced transcripts:

$$\frac{dA_U}{dt} = \frac{G_A}{\prod_i \lambda_{B_iA}} \prod_i \left( \lambda_{B_iA} + \frac{(1 - \lambda_{B_iA})}{1 + \left(\frac{B_i}{B_iA_0}\right)^{n_{B_iA}}} \right) - \beta A_U, \qquad (10)$$

where $\beta$ is the splicing rate. Spliced transcript dynamics are then given by Eq. (11):

$$\frac{dA_s}{dt} = \beta A_U - kA_S, \qquad (11)$$

We simulated stochastic trajectories for 5000 (REP) or 100,000 (CTS) unit time and sampled the expression values from 5000 (REP) or 10,000 (CTS) random timepoints to use as input for both

STICCC and scVelo. These stochastic trajectories were used to collect a distribution of observed trajectories for cells passing through various points in gene expression space. This distribution was used to evaluate the agreement between predicted trajectories from STICCC, scVelo, and the observed trajectories in the same manner as described in the previous section. We also compared the results to those of scVelo when using the dynamical model (Bergen et al, 2020), as well as with another RNA velocity method named veloVI (Gayoso et al, 2024). We compared between methods by calculating the cosine similarity of vectors between paired samples and evaluated statistical significance using the paired Wilcoxon test.

## Basin position estimation

To identify the position of gene expression basins, we calculated the 2D kernel density in the first two principal components of the simulated gene expression data and identified local minima in the space. Due to the random sampling of parameters, some spurious minima far from the visible gene expression clusters were manually removed.

## Processing experimental gene expression data

The scRNA-Seq data for the cell cycle in budding yeast were obtained from a previously published study, which sequenced transcripts from 38,285 budding yeast cells under multiple treatments (Jackson et al, 2020; Data ref: Jackson et al, 2019). In preprocessing, 976 wild-type cells grown in YPD (yeast extract, peptone, glucose) were subset from the complete data and log-normalized. Non-expressed genes were removed from the initial total of 6828, leaving 5623 log counts values per cell, which were transformed to z-scores. The 15 genes chosen for the GRN were based on extensive review of published literature, as detailed in Katebi et al, 2020 and Appendix Table S4.

The scRNA-seq data for U2OS cells were obtained from a previous work which measured gene expression in 1152 U2OS FUCCI cells (Mahdessian et al, 2021; Data ref: Mahdessian et al, 2020). Beginning from TPM data, we log-transformed the data and selected the top 2000 variable genes to use for dimension reduction, as well as assigning cell cycle phases using the Seurat R package (Hao et al, 2024). To create an input GRN for STICCC to use with the U2OS FUCCI cells, we began from a network published in a work comparing the regulatory networks of the mammalian and yeast cell cycle (Medina et al, 2016). We also refined the network manually by comparison to known interactions in KEGG (Kanehisa et al, 2025), as well as identifying homologous genes from the yeast cell cycle network using the Ensembl database (Dyer et al, 2025). In cases where the gene from the network topology was not expressed in the data, or the network topology showed a protein complex (rather than a single gene), we manually selected the appropriate gene by inspection of the gene expression data and known aliases—the full set of genes used for STICCC is available in Appendix Table S5.

Hematopoietic stem cell data were obtained from a previously published study, which generated single-cell qPCR data from triplicates of blood progenitor EML cells treated with either EPO, GM-CSF/IL3, and ATRA, or a mixture of both (Mojtahedi et al, 2016; Data ref: Mojtahedi et al, 2016). The former two treatments were designed to cause differentiation into erythroid and myeloid

cells, respectively. Beginning from the median Qc values across these replicates for 17 genes of interest identified in the previous study, we subtracted the LOD for each gene to obtain log-gene expression values and combined data across timepoints and treatments. Of these genes, several were excluded from the network and expression data as they had low expression and low variance, leaving a final network of nine genes (network topology in Appendix Table S6).

The EMT dataset was obtained from a work published in 2020, which examined EMT across four cancer cell lines and three signaling conditions. ScRNA-seq data were captured at 8 timepoints throughout signal induction and removal. We selected data from A549 cells treated with TGF-β and kept only the first 7 days because few measurements were taken from the later timepoints, and it did not appear that a full reversal of EMT took place (Cook and Vanderhyden, 2020; Data ref: Cook and Vanderhyden, 2020). Gene expression values were log-normalized with a pseudocount of 1. The GRN topology was produced by applying the GRN inference method NetAct to this dataset, resulting in a network of 29 genes (Su et al, 2022) (network topology available on GitHub for STICCC analysis). However, edges to and from the gene ESR1 were removed as no transcripts for it were captured in the measurements, and a further five genes were removed, which had consistently low variance and expression in the profiled cells (E2F2, IRF9, PPARD, SRF, KLF4). For each dataset where STICCC was applied, the sampling radius and gene expression space used as input is listed in Appendix Table S7.

### Benchmarking computational cost

To estimate the runtime at various GRN sizes, we generated inferred GRNs using GENIE3 (Huynh-Thu et al, 2010) and a microarray dataset of A549 cells undergoing EMT obtained from GEO:GSE17708 (Sartor et al, 2010; Data ref: Keshamouni et al, 2009). For input to GENIE3, we first inferred TF activity in the dataset using the R package NetAct; networks were generated by varying the link weight threshold, producing GRNs of sizes between 25 and 1000 edges. Each GRN was supplied to STICCC using the full dataset of 3133 samples, and vector inference was performed 5 times using compute nodes on the Northeastern Explorer cluster, running a CentOS operating system on a dual-socket Intel Xeon E5-2680 v2 CPU. We also benchmarked time costs as a function of dataset size, using simulated datasets of the CTS circuit, which ranged in size from 1000 to 10,000 cells, performing 5 replicates for each size. We evaluated the time cost of vector inference with the R package microbenchmark (Mersmann, 2025). We found the vector inference time appears to scale linearly with the number of cells but does not appear sensitive to GRN size (Appendix Fig. S9).

## Data availability

The datasets and computer code produced in this study are available in the following databases: Modeling and analysis scripts: GitHub (https://github.com/lusystemsbio/sticcc_analysis/). STICCC software package: GitHub (https://github.com/lusystemsbio/sticcc/).

The source data of this paper are collected in the following database record: biostudies:S-SCDT-10_1038-S44320-026-00196-8.

## Peer review information

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

## Acknowledgements

D Ramirez and M Lu are supported by startup funds from Northeastern University, by the National Institute of General Medical Sciences of the National Institutes of Health under Award Number R35GM128717, and by the National Science Foundation under Award Number MCB-2114191. D Ramirez was also supported by a Northeastern University Bioengineering Department Dean's Fellowship. D Ramirez and M Lu acknowledge their affiliation with the Center for Theoretical Biological Physics at Northeastern University and appreciate the support provided by the center.

## Author contributions

**Daniel A Ramirez**: Formal analysis; Validation; Investigation; Visualization; Methodology; Writing—original draft; Writing—review and editing. **Mingyang Lu**: Conceptualization; Data curation; Supervision; Funding acquisition; Investigation; Methodology; Writing—review and editing.

Source data underlying figure panels in this paper may have individual authorship assigned. Where available, figure panel/source data authorship is listed in the following database record: biostudies:S-SCDT-10_1038-S44320-026-00196-8.

## Disclosure and competing interests statement

The authors declare no competing interests.

# Expanded View Figures

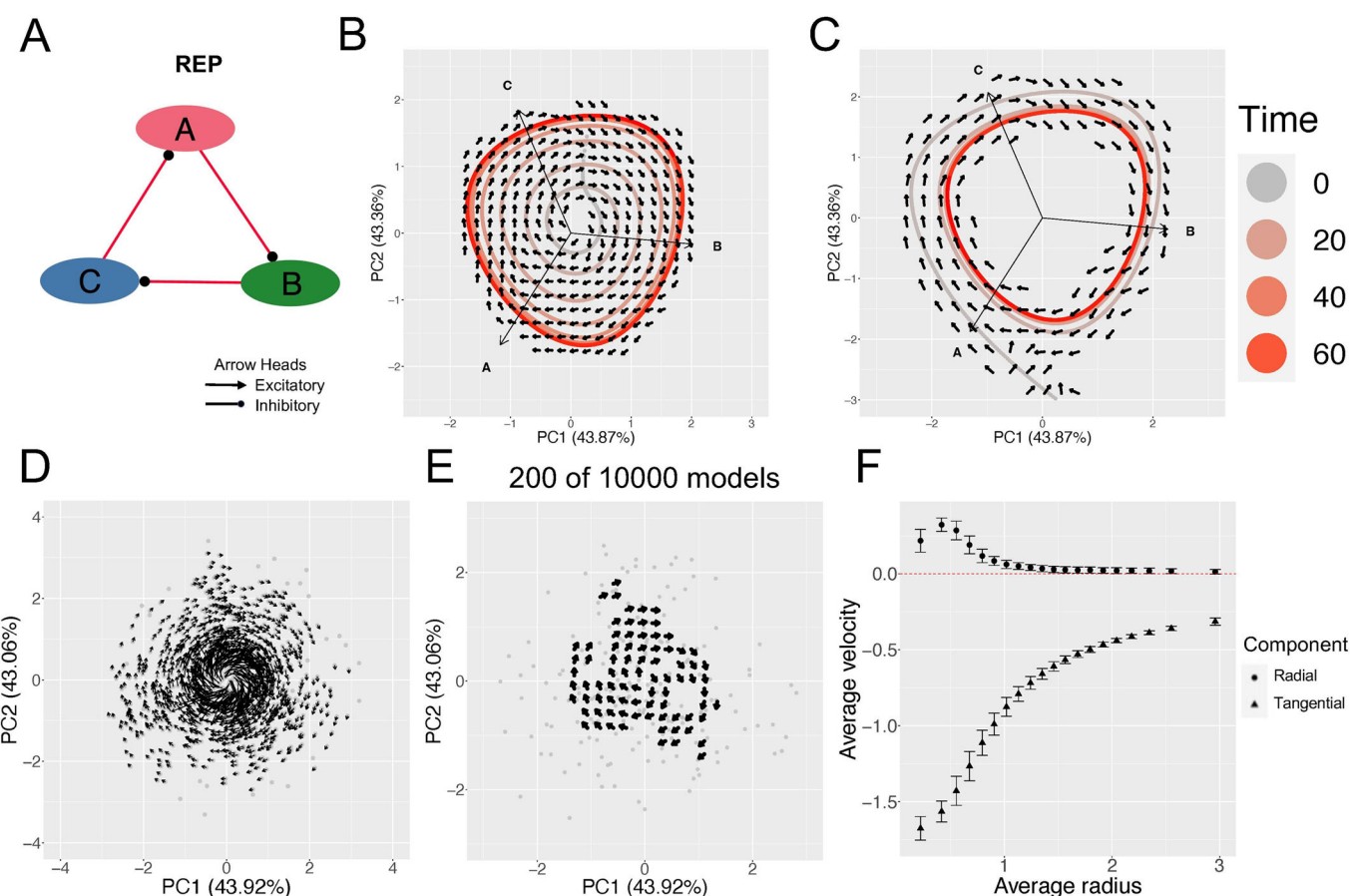

**Figure EV1.  Repressilator (REP) time trajectory simulations and additional analysis.**

(**A**) REP circuit topology diagram. (**B**, **C**) STICCC predictions (vectors shown as arrows) using the gene expression snapshots from the simulated REP trajectories approaching the limit cycle (with later time points indicated in red) from two different initial conditions (with earlier time points indicated in gray). The plots show the projection of the time-series gene expression data onto the first two principal components from the simulated gene expression of an ensemble of 10,000 models. (**D**) Cell-specific outgoing transition vectors v_1 calculated for a simulated gene expression of an ensemble of 10,000 REP models. Shown is a random subset of 1000 cells and their predicted vectors. (**E**) State transition pattern is conserved even with extreme under-sampling. Grid-smoothed vector field for REP circuit is shown after calculating vectors on a subset of 200 out of 10,000 models. (**F**) Mean values of the radial (circles) and tangential (triangles) components of the inferred vectors for cells in various radial bands around the origin of the gene expression space, projected onto the first two principal components ($n = 500$ per $x$ value). The inferred transition vectors are predominantly dominated by the tangential components, suggesting oscillatory state transitions. Error bars are drawn at $+/-$ 1 standard deviation.

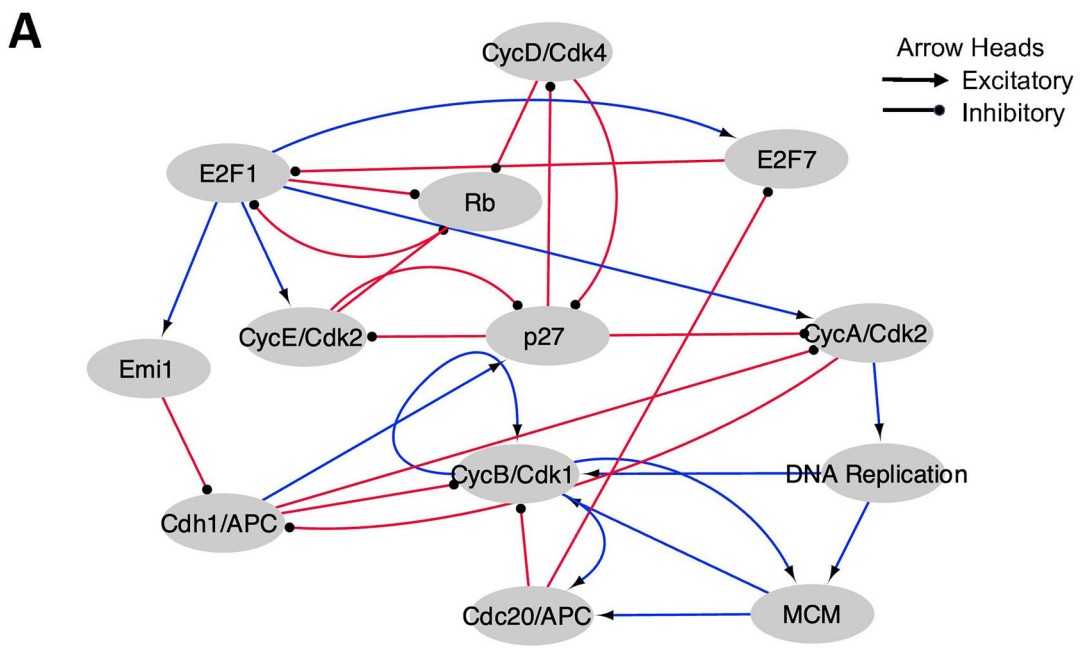

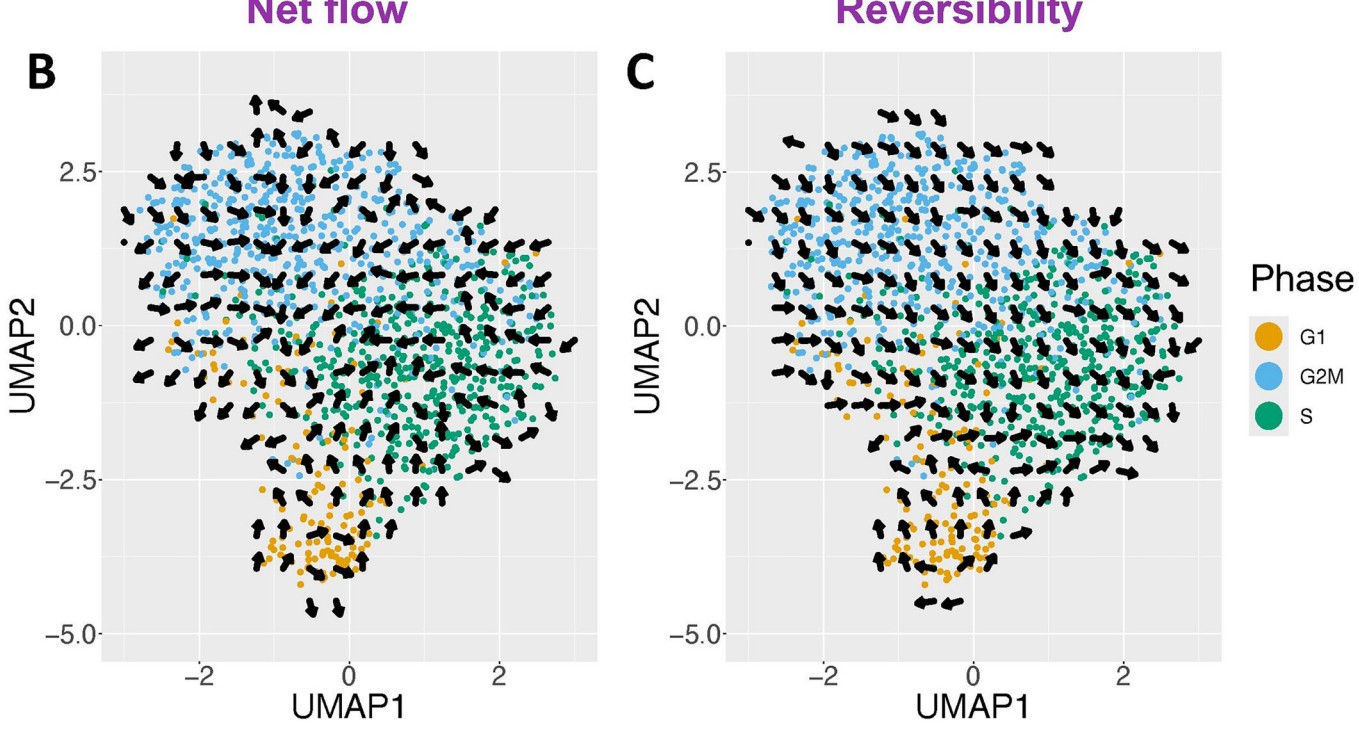

**Figure EV2. STICCC predictions for U2OS cell cycle scRNA-seq data.**

(A) Modified cell cycle circuit topology for mammalian systems. (B) UMAP projection of U2OS cells with STICCC net flow predictions. Points are colored by cell cycle phase. (C) UMAP projection of U2OS cells with STICCC reversibility predictions.

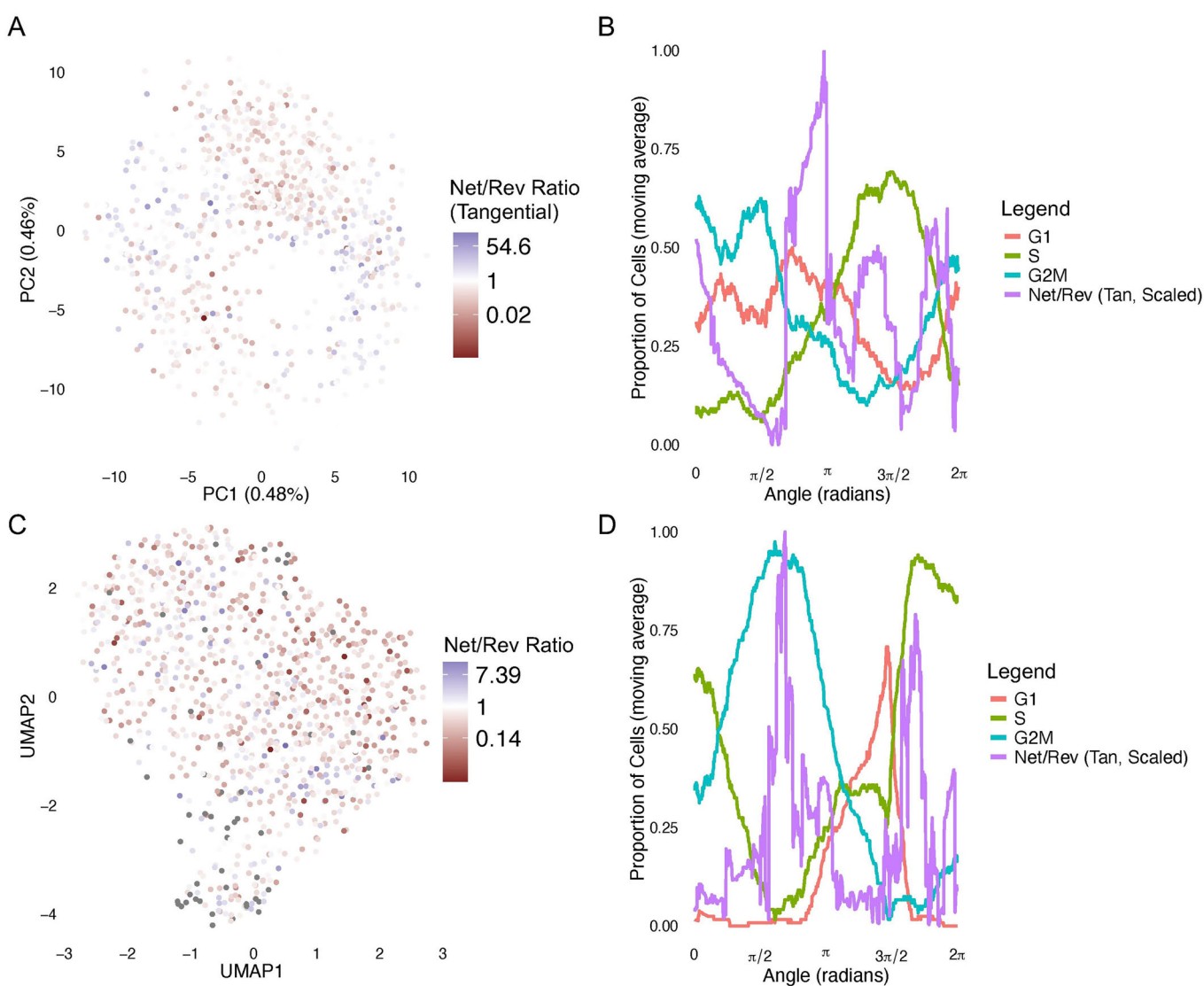

**Figure EV3.  Analysis of STICCC vectors along cell cycle phases.**

(A) PCA projection of yeast scRNA-seq data colored by the ratio of net flow to reversibility magnitude. (B) Line plot showing a moving average (over a 30-cell window) of the proportion of cells in each cell cycle phase as a function of angle from the origin (angles beginning from zero on the positive *x* axis). Purple series shows the ratio of net flow to reversibility magnitude (tangential component only), scaled to a range of 0–1. (C) PCA projection of U2OS scRNA-seq data colored by the ratio of net flow to reversibility magnitude. (D) Line plot showing a moving average (over a 30-cell window) of the proportion of cells in each cell cycle phase as a function of angle from the origin (angles beginning from zero on the positive *x* axis). Purple series shows the ratio of net flow to reversibility magnitude (tangential component only), scaled to a range of 0–1.

    

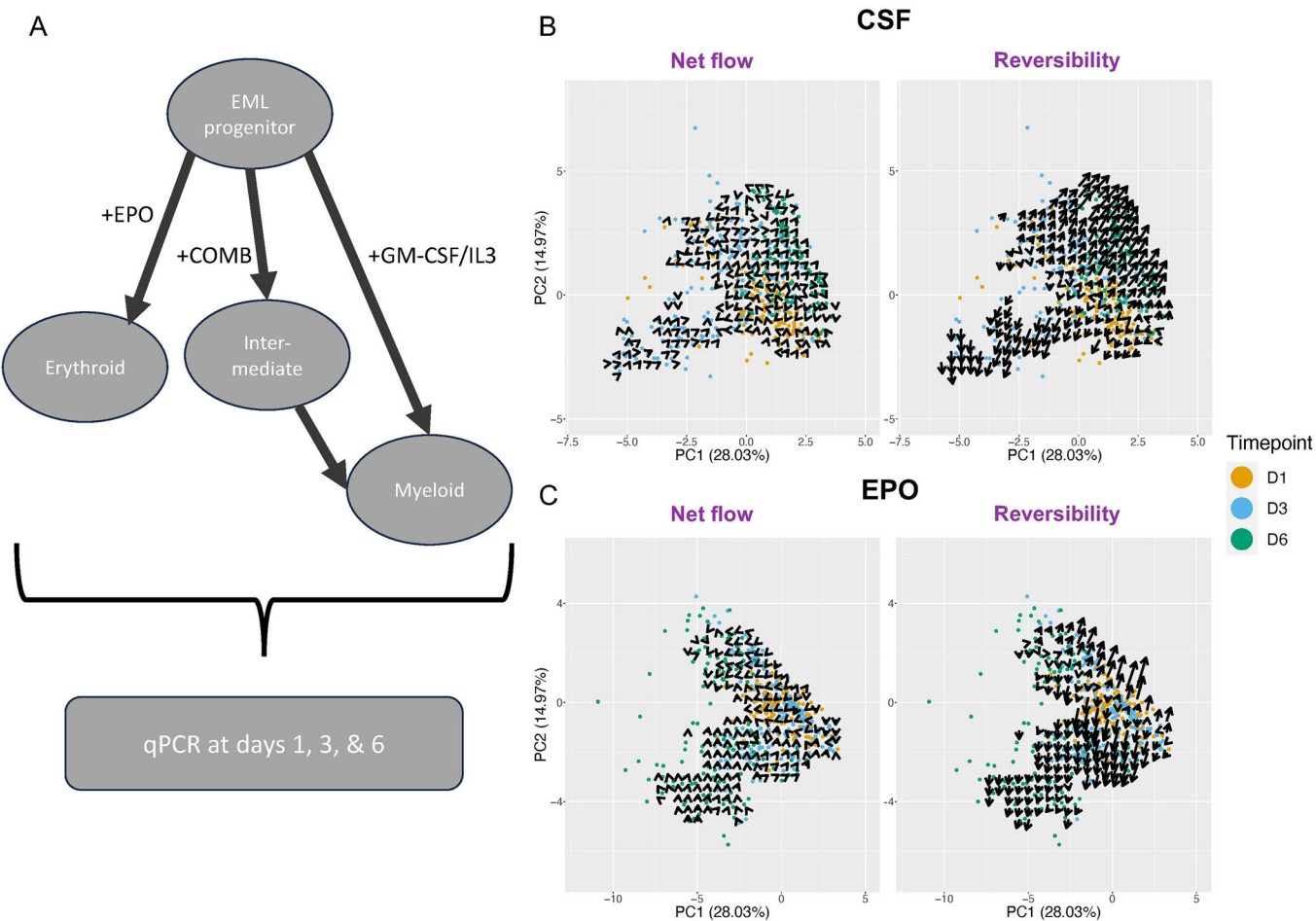

**Figure EV4.  STICCC results for HSC dataset separated by treatment.**

(**A**) Diagram describing the experimental design for HSC dataset: EML progenitor cells were treated with EPO, GM-CSF/IL3, or a combination of both stimuli. (**B**) Net flow and reversibility predictions for HSC cells treated with CSF. Point color denotes timepoint. (**C**) Predictions for HSC cells treated with EPO.

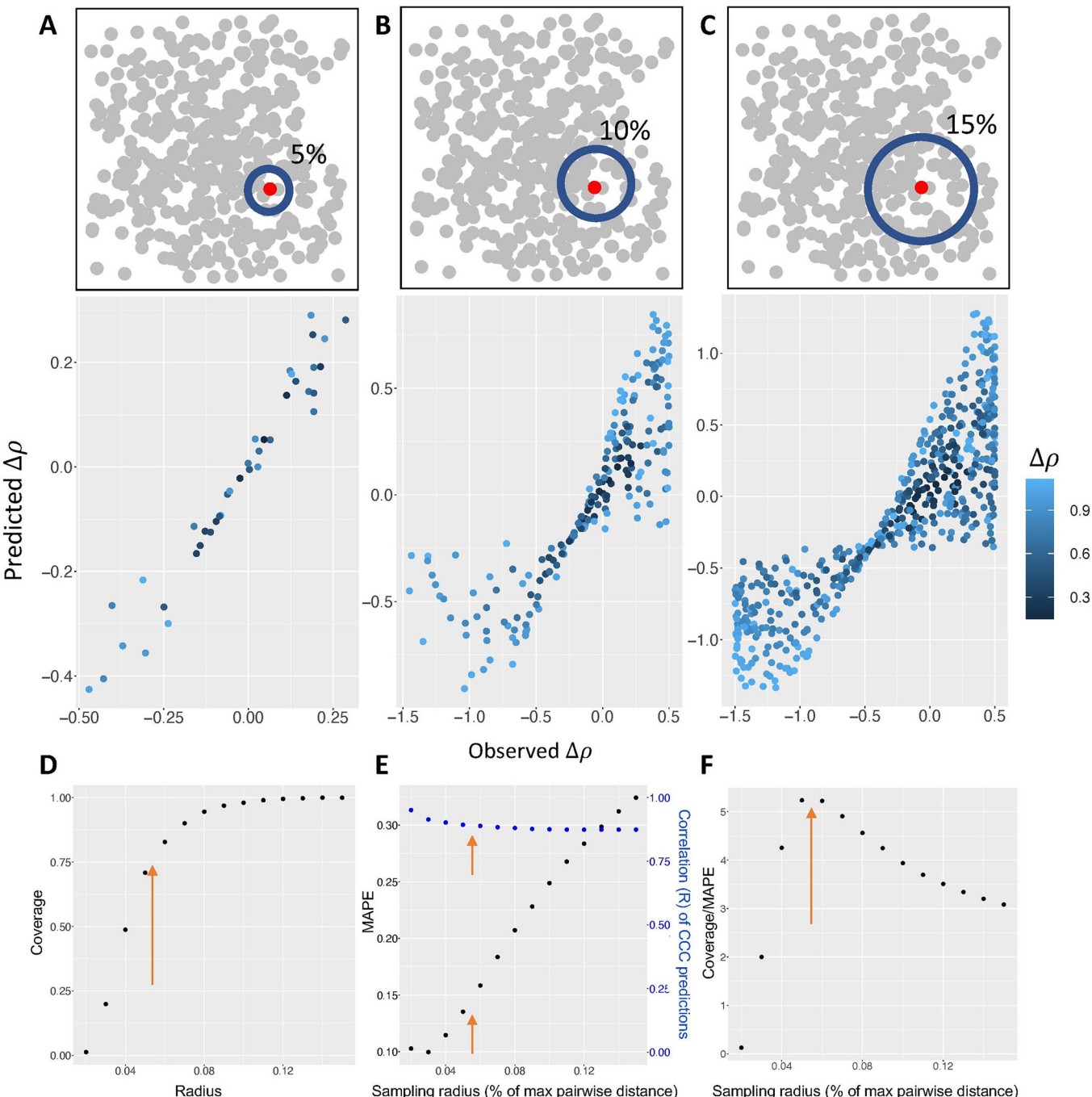

**Figure EV5. Linear regression quality depends on sampling radius.**

(A–C) Example neighborhoods for a sampling radius of 5%, 10%, and 15% of the maximum pairwise Euclidean distance in gene expression between cells, respectively, illustrated by a blue circle around a red center cell. Below, scatterplots show observed and predicted CCC values for neighboring cells. (D) Coverage, i.e., proportion of cells for which a prediction is generated, as a function of sampling radius. (E) Median absolute percent error (MAPE), left axis, and R value of linear regression, right axis, for various sampling radii. (F) Ratio of coverage to MAPE for various sampling radii.

