## [Peer Review File · Molecular Systems Biology]

Dissecting reversible and irreversible single cell state transitions from gene regulatory networks

Daniel Ramirez and Mingyang Lu

Corresponding author(s): Mingyang Lu (m.lu@northeastern.edu)

Review Timeline:

Submission Date:	1st Nov 24
Editorial Decision:	13th Dec 24
Revision Received:	16th Jun 25
Editorial Decision:	29th Jul 25
Revision Received:	17th Nov 25
Editorial Decision:	5th Jan 26
Revision Received:	9th Jan 26
Accepted:	26th Jan 26

Editor: Poonam Bheda

Transaction Report:

13th Dec 2024

Manuscript Number: MSB-2024-12735

Title: Dissecting reversible and irreversible single cell state transitions from gene regulatory networks

Author: Daniel Ramirez

Mingyang Lu

Dear Dr Lu,

Thank you for submitting your work to Molecular Systems Biology. We have now heard back from the referees who agreed to evaluate your manuscript. As you will see below, the reviewers raise important concerns on your work and remain unconvinced that some of the major conclusions are sufficiently supported by the analyses. They thus raise the following major issues:

- limitations of STICCC need to be more explicitly described, in particular the dependency on the on accuracy of GRN used (Reviewers 1, 2, and 3)
- benchmarking of STICCC to RNA velocity methods need to be included to show indeed an advance and perhaps some toning down on the critique of the RNA velocity methods (Reviewers 1 and 3)

In line with the reviewers comments, we would also suggest that improving the manuscript according to the following suggestions would increase the impact of the paper:

- Transition and trajectory inferences should be strengthened on real datasets (Reviewers 1 and 3)
- including recommendations/strategies for helping users to identify most appropriate GRN (Reviewer 3)

If you feel you can satisfactorily address these points and those listed by the referees, you may wish to submit a revised version of your manuscript. If you would like to discuss further the points raised by the referees, I am available to do so via email or video. Let me know if you are interested in this option.

We require:

1) A .docx formatted version of the manuscript text (including legends for main figures, EV figures and tables). Please make sure that the changes are highlighted to be clearly visible. Alternatively you may choose to submit your manuscript as a LaTeX file.

4) A .docx formatted letter INCLUDING the reviewers' reports and your detailed point-by-point responses to their comments. As part of the EMBO Press transparent editorial process, the point-by-point response is part of the Peer Review File (PRF), which will be published alongside your paper.

5) A complete author checklist, which you can download from our author guidelines (<https://www.embopress.org/page/journal/17574684/authorguide#submissionofrevisions>). Please insert information in the checklist that is also reflected in the manuscript. The completed author checklist will also be part of the PRF.

6) Please note that all corresponding authors are required to supply an ORCID ID for their name upon submission of a revised manuscript.

7) It is mandatory to include a 'Data Availability' section after the Materials and Methods. Before submitting your revision, primary datasets produced in this study need to be deposited in an appropriate public database, and the accession numbers and database listed under 'Data Availability'. Please remember to provide a reviewer password if the datasets are not yet public (see <https://www.embopress.org/page/journal/17574684/authorguide#dataavailability>).

In case you have no data that requires deposition in a public database, please state so in this section. Note that the Data Availability Section is restricted to new primary data that are part of this study. This study includes no data deposited in external repositories.

8) All Materials and Methods need to be described in the main text using our 'Structured Methods' format, which is required for all research articles. According to this format, the Methods section includes a Reagents and Tools Table (listing key reagents, experimental models, software and relevant equipment and including their sources and relevant identifiers) followed by a Methods and Protocols section describing the methods using a step-by-step protocol format. The aim is to facilitate adoption of the methodologies across labs. Please upload the Reagents and Tools table as a separate document when submitting your revised manuscript. More information on how to adhere to this format as well as a downloadable template (.docx) for the Reagents and Tools Table can be found in our author guidelines:
<https://www.embopress.org/page/journal/17444292/authorguide#structuredmethods>

An example of a Method paper with Structured Methods can be found here:
<https://www.embopress.org/doi/10.15252/msb.20178071>.

9) For data quantification: please specify the name of the statistical test used to generate error bars and P values, the number (n) of independent experiments (specify technical or biological replicates) underlying each data point and the test used to calculate p-values in each figure legend. The figure legends should contain a basic description of n, P and the test applied. Graphs must include a description of the bars and the error bars (s.d., s.e.m.). Please provide exact p values.

10) Our journal encourages inclusion of *data citations in the reference list* to directly cite datasets that were re-used and obtained from public databases. Data citations in the article text are distinct from normal bibliographical citations and should directly link to the database records from which the data can be accessed. In the main text, data citations are formatted as follows: "Data ref: Smith et al, 2001" or "Data ref: NCBI Sequence Read Archive PRJNA342805, 2017". In the Reference list, data citations must be labeled with "[DATASET]". A data reference must provide the database name, accession number/identifiers and a resolvable link to the landing page from which the data can be accessed at the end of the reference. Further instructions are available at .

11) We replaced Supplementary Information with Expanded View (EV) Figures and Tables that are collapsible/expandable online. A maximum of 5 EV Figures can be typeset. EV Figures should be cited as 'Figure EV1, Figure EV2" etc... in the text and their respective legends should be included in the main text after the legends of regular figures.

<https://www.embopress.org/page/journal/17574684/authorguide#expandedview>

13) Author contributions: CRediT has replaced the traditional author contributions section because it offers a systematic machine readable author contributions format that allows for more effective research assessment. Please remove the Authors Contributions from the manuscript and use the free text boxes beneath each contributing author's name in our system to add specific details on the author's contribution. More information is available in our guide to authors.

Please also suggest a striking image or visual abstract to illustrate your article as a PNG file 550 px wide x 300-600 px high. Share synopsis text and image, as well as eTOC:

Please note that these would be the final versions and changes during proofing are usually not allowed

16) As part of the EMBO Publications transparent editorial process initiative (see our policy here: https://www.embopress.org/transparent-process#Review_Process), Molecular Systems Biology will publish online a Peer Review File (PRF) to accompany accepted manuscripts.

In the event of acceptance, this file will be published in conjunction with your paper and will include the anonymous referee reports, your point-by-point response and all pertinent correspondence relating to the manuscript. Let us know whether you agree with the publication of the PRF and as here, if you want to remove or not any figures from it prior to publication. Please note that the Authors checklist will be published at the end of the PRF.

Molecular Systems Biology has a "scooping protection" policy, whereby similar findings that are published by others during review or revision are not a criterion for rejection. Should you decide to submit a revised version, I do ask that you get in touch after three months if you have not completed it, to update us on the status.

I look forward to receiving your revised manuscript.

Yours sincerely,

Poonam Bheda, PhD
Scientific Editor
Molecular Systems Biology

Reviewer #1:

In this article, the authors presented a computational method designed to infer state transition dynamics from snapshot single-cell data. They applied this method to both simulated and experimental single-cell datasets to predict reversible and irreversible state transitions based on gene expression profiles. Furthermore, through edge sensitivity analysis in STICCC, the authors demonstrated how specific network interactions influence reversibility and state transitions. While the approach is insightful, I have the following comments and suggestions regarding the implementation of the methods:

1. In the initial step of the STICCC method, the authors calculate transition probabilities based on the correlation between the expression profile of a given cell and those of its nearest neighbors. However, an alternative approach, such as using the diffusion operator, could have been employed to estimate these probabilities. The rationale for choosing cross-cell correlation over other methods, such as diffusion-based approaches, is unclear and should be explicitly clarified.
2. The authors utilize two transition vectors-incoming and outgoing-and compute the net effect to generate a vector field. The authors provided a comparison with the RNA velocity method in the discussion section. The authors should discuss how their approach enhances the estimation of state transitions compared to RNA velocity and whether it addresses any specific limitations of the latter.
3. The comments on RNA velocity and related methods are largely incorrect. Their criticisms are not justified. In Line 73-75, the "unique advantages" claimed by the authors can be achieved with the existing RNA velocity-based methods.
4. In Fig. 2, the authors depict overall transition patterns using colored arrows. It is unclear what criteria were used to determine the placement and orientation of these arrows. Were the arrows positioned for illustrative purposes only, or do they reflect an aggregated transition pattern derived from the directions of individual points in the state space?
5. In the demonstrations of transition vectors with different motifs (Fig. 2), it would be more informative to include the corresponding gene expression profiles alongside the vectors in the PCA space. This would provide additional context and help in interpreting the relationship between the vector patterns and gene expression dynamics.
6. The authors claim that in the case of a coupled bistable motif with negative feedback, STICCC can infer oscillatory vectors along with reversible transitions between the basins in the net flow. However, it is not clear from Fig. 2H the positions of the two basins. Providing additional annotations or explanations to identify these basins would enhance the clarity of this figure.
7. While the authors demonstrate that STICCC can effectively detect signal-induced state transitions from snapshot data, it would be helpful to visualize the positions of the basins and saddle points as a function of signal strength in a single plot. This additional analysis would provide a more comprehensive understanding of the relationship between signal strength and state transition dynamics.
8. The authors identify a circular vector field in the scRNA-seq dataset of budding yeast, representing the cell cycle. Since the cell cycle involves distinct checkpoints where cells decide to transition from one state to another, it would be interesting to explore whether STICCC can pinpoint the positions of these checkpoints.
9. The authors acknowledge the computational cost of the method and note the need for further improvements to predict reversibility in high-resolution time-series scRNA-seq data. However, the authors should provide a more comprehensive discussion of the algorithm's limitations, including specific challenges or potential sources of error.

Reviewer #2:

This manuscript describes a method called STICCC, using scRNAs-seq data to predict reversible and irreversible cell state transitions. Similar method comparable to STICCC is RNA velocity. But the authors of this manuscript present their method with GRN information is incorporated and claim that their method provides complementary insights into the gene regulation of cell state transitions and offers an approach for identifying key regulators within GRNs. STICCC looks indeed a powerful tool for studying cell state transitions. Several strengths include its ability to predict both reversible and irreversible transitions and its robustness to noise, which is because it uses a local distribution of samples to compute each vector.

Major points

(1) STICCC leverages time delays in gene expression between regulators and their targets to infer past and future states. This makes the method sensitive to the inaccurate representation of regulatory relationships between genes. Errors in network topology, missing interactions, or incorrect classifications of activation/inhibition relationships can all propagate into the STICCC predictions, leading to misleading interpretations of cell state transitions. Without accurate GRN, the calculated CCC values and subsequent vector predictions would be unreliable. Thus, the most significant limitation of STICCC is its reliance on a known and accurate GRN as input.

(2) The computational cost may make the scaling up of STICCC a problem. The calculation of pairwise CCC values between cells involves computationally intensive task, which leads to a computational complexity that scales quadratically with the number of cells in the dataset.

(3) The "validation" of the STICCC's predictions is primarily depended upon simulated data with a limited number of experimental datasets with relatively coarse time resolution. PCA is used for dimensional reduction and visualize cells, which could be okay in the case of data with coarse time resolution. For high-resolution time-series scRNA-seq data, PCA may not be able to do a good job separating cells in different states to separable groups.

(4) The vector field plot generated by STICCC is difficult to interpret for users who are not familiar with the tool. There is no quantitative approach provided to assist the effort. The task is like looking at an X-ray scatter plot and trying to figure out the 3D structure of a protein.

Minor point:

(1) Github page - Installation command line:
devtools::install_github("lusystemsbio/viccc.git")
might should be
devtools::install_github("lusystemsbio/viccc")

(2) Source code: running error -
> # grid-based smoothing of velocities
> stic <- computeGridVectors(stic)
Error in grid.df[, c("dx", "dy")] : incorrect number of dimensions

Reviewer #3:

The article presents a novel approach for RNA velocity inference using gene regulatory networks and transcriptomics data, eliminating the reliance on spliced and unspliced counts required by existing methods. The method assumes that changes in regulators' activity or expression precede changes in their target genes' expression. Thus, it leverages gene regulatory interactions and gene expression data to infer vector fields that represent reversible and irreversible state transitions. These vectors are constructed using cross-cell correlations between the expression levels of regulators in a cell and their targets in the cell's neighbors.

The authors evaluate their method on ten simulated datasets and three real datasets, encompassing cell counts ranging from 200 to 10,000. Results demonstrate the method's ability to characterize cell state transitions, infer trajectories, and identify the reversibility of these transitions. Overall, the manuscript is well-written, and the method is both innovative and supported by promising findings. However, the evaluation could be expanded to enhance the manuscript's impact. Below are a few suggestions to strengthen the work:

1. The accuracy of cell state transition inference using this method seems to rely heavily on the selection of the input gene regulatory network (GRN). However, prior knowledge of GRNs is often limited, varies across experimental conditions, and is highly cell type-specific. Do the authors have any recommendations or strategies to help users identify or select the most appropriate GRN for their analyses?
2. The current validation is restricted to simulated datasets and small-scale real experiments, which typically involve only one cell type and a single regulatory network. How does the method perform on more complex datasets that include multiple cell types or developmental stages controlled by distinct GRNs?

3. The regulatory networks used in the current evaluation appear relatively small. It would be valuable for the authors to assess how the complexity and size of the GRN influence the method's performance.
4. To better demonstrate the performance of the proposed method, the authors should include comparisons with existing RNA velocity inference methods, such as CellRank and VeloCycle.
5. The trajectory inference analysis raises some concerns. GRNs are often inferred computationally from scRNA-seq data, and many methods rely on known trajectories or cell orderings obtained through pseudotime inference (e.g., PMID: 34020546). Therefore, the claim of improving trajectory inference might be less compelling without further evidence. The authors could strengthen their argument by providing additional analyses on real single-cell datasets and benchmarking their method against established pseudotime trajectory inference approaches.

Response letter

We are thankful to the referees for their careful reading of the manuscript and for their thoughtful and constructive comments and suggestions. In revising the manuscript, we made sure to address all the issues enumerated by the referees. Doing so led to significant improvement of the article and made the descriptions clearer and results sounder. Below we have detailed all the changes made (our response in red in the letter and changes in red in the revised manuscript) point-by-point according to the referees' input (in black).

Reviewer #1:

In this article, the authors presented a computational method designed to infer state transition dynamics from snapshot single-cell data. They applied this method to both simulated and experimental single-cell datasets to predict reversible and irreversible state transitions based on gene expression profiles. Furthermore, through edge sensitivity analysis in STICCC, the authors demonstrated how specific network interactions influence reversibility and state transitions. While the approach is insightful, I have the following comments and suggestions regarding the implementation of the methods:

1. In the initial step of the STICCC method, the authors calculate transition probabilities based on the correlation between the expression profile of a given cell and those of its nearest neighbors. However, an alternative approach, such as using the diffusion operator, could have been employed to estimate these probabilities. The rationale for choosing cross-cell correlation over other methods, such as diffusion-based approaches, is unclear and should be explicitly clarified.

We thank the reviewer for calling attention to alternative approaches. We believe that cross-cell correlation, compared to diffusion, has the advantage that it 1) directly incorporates causal biological prior knowledge in describing the likelihood of transitions, and 2) emphasizes the local gene expression landscape. Whereas a diffusion-based approach would infer reversible transition probabilities based on a global gene expression manifold, the approach we describe leverages nearby gene expression states and known regulatory interactions to predict directional state transitions. We have updated the Introduction section of the manuscript (lines 68-71, 76-80) to highlight this distinction.

2. The authors utilize two transition vectors-incoming and outgoing-and compute the net effect to generate a vector field. The authors provided a comparison with the RNA velocity method in the discussion section. The authors should discuss how their approach enhances the estimation of state transitions compared to RNA velocity and whether it addresses any specific limitations of the latter.

We thank the reviewer for encouraging clarity on this important point. We have modified the manuscript to include a more detailed comparison and discussion of RNA velocity. Specifically, we expanded on our discussion (lines 400-417) and emphasize the value of STICCC's ability to directly address reversibility of transitions, capturing the behavior of multi-stable systems in a way that RNA velocity cannot. We also provide additional examples directly comparing the results of RNA velocity and STICCC on simulated trajectories in Figure 3 (lines 254-263).

Finally, although we are unable to perform RNA velocity analysis on some of the experimental datasets shown (because yeast do not have the same RNA processing machinery that the method exploits, and the HSC dataset comes from qPCR measurements), we obtained a dataset describing the cell cycle in human U2OS cells which has been used in prior RNA velocity analyses, finding a similar oscillatory structure as was previously reported (Fig. EV2, lines 299-315). We also analyzed the relative speed of phase changes via the ratio of net flow to reversibility vectors (Fig. EV3), finding slowdowns ahead of phase changes followed by sharp acceleration, which we hypothesize could be associated with cell cycle checkpoints being reached and subsequently overcome.

3. The comments on RNA velocity and related methods are largely incorrect. Their criticisms are not justified. In Line 73-75, the "unique advantages" claimed by the authors can be achieved with the existing RNA velocity-based methods.

We thank the reviewer for bringing this to our attention. We have rewritten the relevant section to better contextualize our work with respect to other related methods. In particular, we highlight that our method explicitly identifies reversible and irreversible transitions, uses a GRN to inform transition probabilities, and strongly weights the local gene expression landscape. To the authors' knowledge, there is no existing RNA velocity-based method which incorporates all of these elements.

4. In Fig. 2, the authors depict overall transition patterns using colored arrows. It is unclear what criteria were used to determine the placement and orientation of these arrows. Were the arrows positioned for illustrative purposes only, or do they reflect an aggregated transition pattern derived from the directions of individual points in the state space?

The arrows drawn in Fig. 2 are illustrative and intended to summarize the overall patterns visible from the individual inferred vectors. We have clarified this point in the relevant figure legend.

5. In the demonstrations of transition vectors with different motifs (Fig. 2), it would be more informative to include the corresponding gene expression profiles alongside the vectors in the PCA space. This would provide additional context and help in interpreting the relationship between the vector patterns and gene expression dynamics.

We agree that the gene expression profiles are a useful complement to interpreting these plots, but feel that incorporating further information into the figure may make it difficult to read. We have added supplementary figures showing gene expression distributions, now present in Appendix Fig. S1.

6. The authors claim that in the case of a coupled bistable motif with negative feedback, STICCC can infer oscillatory vectors along with reversible transitions between the basins in the net flow. However, it is not clear from Fig. 2H the positions of the two basins. Providing additional annotations or explanations to identify these basins would enhance the clarity of this figure.

We thank the reviewer for pointing out this important detail. We have estimated the positions of basins based on the distribution of simulated gene expression data and indicated these with red

dots in Fig. 2, as well as independently showing the density distribution and basin positions in Appendix Fig. S1.

7. While the authors demonstrate that STICCC can effectively detect signal-induced state transitions from snapshot data, it would be helpful to visualize the positions of the basins and saddle points as a function of signal strength in a single plot. This additional analysis would provide a more comprehensive understanding of the relationship between signal strength and state transition dynamics.

We agree with the reviewer's valuable suggestion and have incorporated a new Appendix Figure S4 which shows the movement of the gene expression basins over time, confirming that they are highly correlated with the signal strength in this example, although we note that this relationship may not be so strong in larger GRNs.

8. The authors identify a circular vector field in the scRNA-seq dataset of budding yeast, representing the cell cycle. Since the cell cycle involves distinct checkpoints where cells decide to transition from one state to another, it would be interesting to explore whether STICCC can pinpoint the positions of these checkpoints.

We thank the reviewer for pointing out this opportunity to further analyze the data. We have added a new Figure EV3, which examines the magnitude of net flow and reversibility vectors throughout the cell cycle. While we note the cell cycle phases are not very well separated by the projection and overlap substantially, we identify local minima and maxima of the ratio between net flow and reversibility, with minima possibly indicating the presence of regulatory checkpoints where additional conditions must be met before proceeding, and maxima indicating parts of the cell cycle which proceed more automatically. These minima appear generally between phase transitions, i.e., near the region in PCA space where a majority of cells in G1 gives way to cells in S, likewise for S into G2M, etc.

9. The authors acknowledge the computational cost of the method and note the need for further improvements to predict reversibility in high-resolution time-series scRNA-seq data. However, the authors should provide a more comprehensive discussion of the algorithm's limitations, including specific challenges or potential sources of error.

We appreciate the reviewer's comment and have revised the Discussion to include a more thorough description of limitations and possible sources of error.

Reviewer #2:

This manuscript describes a method called STICCC, using scRNAs-seq data to predict reversible and irreversible cell state transitions. Similar method comparable to STICCC is RNA velocity. But the authors of this manuscript present their method with GRN information is incorporated and claim that their method provides complementary insights into the gene regulation of cell

state transitions and offers an approach for identifying key regulators within GRNs. STICCC looks indeed a powerful tool for studying cell state transitions. Several strengths include its ability to predict both reversible and irreversible transitions and its robustness to noise, which is because it uses a local distribution of samples to compute each vector.

Major points

(1) STICCC leverages time delays in gene expression between regulators and their targets to infer past and future states. This makes the method sensitive to the inaccurate representation of regulatory relationships between genes. Errors in network topology, missing interactions, or incorrect classifications of activation/inhibition relationships can all propagate into the STICCC predictions, leading to misleading interpretations of cell state transitions. Without accurate GRN, the calculated CCC values and subsequent vector predictions would be unreliable. Thus, the most significant limitation of STICCC is its reliance on a known and accurate GRN as input.

We thank the reviewer for pointing out this important point. We have revised the Discussion section to specifically highlight the importance of an accurate input GRN. We also note some degree of robustness to incorrect topologies in our analysis of simulated data, available in Appendix Figure S5.

(2) The computational cost may make the scaling up of STICCC a problem. The calculation of pairwise CCC values between cells involves computationally intensive task, which leads to a computational complexity that scales quadratically with the number of cells in the dataset.

We agree with the reviewer's concern about scalability and have made some effort to show the method is applicable in practical scenarios. Namely, we show that the predicted vectors are robust to significant down-sampling, showing agreement in results with as few as 200 simulated cells from an initial dataset of 10,000 (Figure EV1). We also have added an analysis of computational time costs (Appendix Figure S7). While in this work, we did not undertake to further optimize the algorithm, we have discussed some possible approaches to do so in the future in our Discussion.

(3) The "validation" of the STICCC's predictions is primarily depended upon simulated data with a limited number of experimental datasets with relatively coarse time resolution. PCA is used for dimensional reduction and visualize cells, which could be okay in the case of data with coarse time resolution. For high-resolution time-series scRNA-seq data, PCA may not be able to do a good job separating cells in different states to separable groups.

We thank the reviewer for pointing out this detail. We believe that PCA is a good choice of projection for many datasets because it preserves local and global distances, and because distance in PCA space is linear. However, we have modified STICCC to support the use of tSNE or UMAP projections, where nearest neighbors are still obtained from principal components (or the full gene expression space), but vectors can be output in any of the above embedding dimensions. We employ this approach in a new example describing the cell cycle in human U2OS cells, where we predict vectors in UMAP space due to the relatively large separation between G1 and other phases in PCA space (Figure EV2).

(4) The vector field plot generated by STICCC is difficult to interpret for users who are not familiar with the tool. There is no quantitative approach provided to assist the effort. The task is like looking at an X-ray scatter plot and trying to figure out the 3D structure of a protein.

We thank the reviewer for pointing out this limitation. We believe that a concise quantitative summary of the circuit behavior is difficult to obtain and, as with similar methods such as RNA velocity, some interpretation of the results must be done simply at the level of visualization. However, in the revision we have incorporated certain quantitative analysis of the results, including examining ratios of net flow/reversibility in various subsets of the data (lines 309-315, Figure EV3); measuring the median change in magnitude of net flow and reversibility with respect to changes in the input GRN (Appendix Figure S5); and grid-based smoothing to mitigate noise that might be present in single-cell vectors. We believe that further development of quantitative analysis tools is possible but outside the scope of this work.

Minor point:

(1) Github page - Installation command line:
devtools::install_github("lusystemsbio/viccc.git")
might should be
devtools::install_github("lusystemsbio/viccc")

We thank the reviewer for pointing this out and have remedied this typographical error.

(2) Source code: running error -
> # grid-based smoothing of velocities
> stic <- computeGridVectors(stic)
Error in grid.df[, c("dx", "dy")] : incorrect number of dimensions

We thank the reviewer for trying out our tool, and for pointing out this issue. Unfortunately, we have not been able to replicate this particular error, but it may be the case that the version on GitHub was not up to date at the time the reviewer encountered the bug. We have tested the published code and vignettes and can report they appear to be working at this time.

Reviewer #3:

The article presents a novel approach for RNA velocity inference using gene regulatory networks and transcriptomics data, eliminating the reliance on spliced and unspliced counts required by existing methods. The method assumes that changes in regulators' activity or expression precede changes in their target genes' expression. Thus, it leverages gene regulatory interactions and gene expression data to infer vector fields that represent reversible and irreversible state transitions. These vectors are constructed using cross-cell correlations between the expression levels of regulators in a cell and their targets in the cell's neighbors.

The authors evaluate their method on ten simulated datasets and three real datasets, encompassing cell counts ranging from 200 to 10,000. Results demonstrate the method's ability to characterize cell state transitions, infer trajectories, and identify the reversibility of these

transitions. Overall, the manuscript is well-written, and the method is both innovative and supported by promising findings. However, the evaluation could be expanded to enhance the manuscript's impact. Below are a few suggestions to strengthen the work:

1. The accuracy of cell state transition inference using this method seems to rely heavily on the selection of the input gene regulatory network (GRN). However, prior knowledge of GRNs is often limited, varies across experimental conditions, and is highly cell type-specific. Do the authors have any recommendations or strategies to help users identify or select the most appropriate GRN for their analyses?

We thank the reviewer for pointing out this important aspect of our work. We agree that the accuracy and quality of the input GRN is a key determinant of the algorithm's success. We have updated our Discussion section with an explanation of how GRN inputs affect the method, as well as some suggested approaches for bootstrapping a GRN from data where a manually curated GRN is not available.

2. The current validation is restricted to simulated datasets and small-scale real experiments, which typically involve only one cell type and a single regulatory network. How does the method perform on more complex datasets that include multiple cell types or developmental stages controlled by distinct GRNs?

We thank the reviewer for the insightful question. We recognize that the input requirement of a single GRN implicitly assumes a shared regulation of all cell types present in the dataset; implicitly, the method assumes that if multiple cell types are present, this is due to multistability inherent in the GRN. We believe our results on simulated and experimental data show that this is a reasonable assumption, and we have added some further discussion on the limitations of the method relating to this point (lines 431-444).

3. The regulatory networks used in the current evaluation appear relatively small. It would be valuable for the authors to assess how the complexity and size of the GRN influence the method's performance.

We thank the reviewer for pointing this out. We expect that as GRN size increases, each edge will have a smaller influence on the resulting vector field, creating a situation where a large and low-confidence network may produce spurious results that appear insensitive to small changes in the topology. However, we note that our analysis in the manuscript covers GRNs between ~3-30 nodes, which we believe is an adequate scale to capture regulatory dynamics across a wide variety of systems. We have also added an analysis of the computational time cost as a function of network and dataset size (Appendix Figure S7), finding that for a fixed number of cells, the time cost is approximately constant with respect to GRN size, and the method scales approximately linearly with increasing numbers of cells.

4. To better demonstrate the performance of the proposed method, the authors should include comparisons with existing RNA velocity inference methods, such as CellRank and VeloCycle.

We thank the reviewer for the suggestion. We have added specific comparisons to RNA velocity method scvelo (used in CellRank) in Figure 3 (lines 299-315) for both simulated and experimental data.

5. The trajectory inference analysis raises some concerns. GRNs are often inferred computationally from scRNA-seq data, and many methods rely on known trajectories or cell orderings obtained through pseudotime inference (e.g., PMID: 34020546). Therefore, the claim of improving trajectory inference might be less compelling without further evidence. The authors could strengthen their argument by providing additional analyses on real single-cell datasets and benchmarking their method against established pseudotime trajectory inference approaches.

We thank the reviewer for addressing this point where our manuscript was not clear. The approach we propose in this manuscript is not intended to perform trajectory inference or refine the results of existing trajectory inference methods, but rather to describe the likelihood and directionality of cell state transitions along the gene expression space. We have added specific comparisons to RNA velocity described above and will consider in future work to investigate the relationship between STICCC predictions and trajectories from, e.g., pseudotime methods.

29th Jul 2025

Manuscript Number: MSB-2024-12735R

Title: Dissecting reversible and irreversible single cell state transitions from gene regulatory networks

Author: Daniel Ramirez

Mingyang Lu

Dear Dr Lu,

Thank you again for submitting your revised work to Molecular Systems Biology. We have now heard back from two of the original three reviewers who evaluated your study, as well as a new reviewer, Reviewer #4. As you will see below, the reviewers are supportive on the utility of the method, and they also confirm that the code functions fine. However, Reviewer #4 finds that the revisions to address previous comments on the reliance on the features and quality of the input GRN as well as comparison to RNA velocity methods rather insufficient. Considering the goal of the manuscript is a method that will hopefully be adopted by the community, editorially we would encourage you to address these comments with additional analyses and clearer discussion on the limitations of the analyses, such as GRN features that were tested (e.g. how many nodes/edges) in the main text (not just the Methods) and how they might affect the outcomes. Please let me know in case you would like to discuss in further detail any of the any of the reviewer comments, I would be happy to schedule a call.

We remind you that we have the following formatting requirements:

1) A .docx formatted version of the manuscript text (including legends for main figures, EV figures and tables). Please make sure that the changes are highlighted to be clearly visible. Alternatively you may choose to submit your manuscript as a LaTeX file.

4) A .docx formatted letter INCLUDING the reviewers' reports and your detailed point-by-point responses to their comments. As part of the EMBO Press transparent editorial process, the point-by-point response is part of the Peer Review File (PRF), which will be published alongside your paper.

5) A complete author checklist, which you can download from our author guidelines (<https://www.embopress.org/page/journal/17574684/authorguide#submissionofrevisions>). Please insert information in the checklist that is also reflected in the manuscript. The completed author checklist will also be part of the PRF.

6) Please note that all corresponding authors are required to supply an ORCID ID for their name upon submission of a revised manuscript.

7) It is mandatory to include a 'Data Availability' section after the Materials and Methods. Before submitting your revision, primary datasets produced in this study need to be deposited in an appropriate public database, and the accession numbers and database listed under 'Data Availability'. Please remember to provide a reviewer password if the datasets are not yet public (see <https://www.embopress.org/page/journal/17574684/authorguide#dataavailability>).

In case you have no data that requires deposition in a public database, please state so in this section as follows: "This study includes no data deposited in external repositories". Note that the Data Availability Section is restricted to new primary data that are part of this study.

8) All Materials and Methods need to be described in the main text using our 'Structured Methods' format, which is required for all research articles. According to this format, the Methods section includes a Reagents and Tools Table (listing key reagents, experimental models, software and relevant equipment and including their sources and relevant identifiers) followed by a Methods and Protocols section describing the methods using a step-by-step protocol format. The aim is to facilitate adoption of the methodologies across labs. Please upload the Reagents and Tools table as a separate document when submitting your revised manuscript. More information on how to adhere to this format as well as a downloadable template (.docx) for the Reagents and Tools Table can be found in our author guidelines:

<https://www.embopress.org/page/journal/17444292/authorguide#structuredmethods>

An example of a Method paper with Structured Methods can be found here:
<https://www.embopress.org/doi/10.15252/msb.20178071>.

9) For data quantification: please specify the name of the statistical test used to generate error bars and p-values, the number (n) of independent experiments (specify technical or biological replicates) underlying each data point and the test used to calculate p-values in each figure legend. The figure legends should contain a basic description of n, p-values and the test applied. Graphs must include a description of the bars and the error bars (s.d., s.e.m.). Please provide exact p-values (in either the figure or figure legend).

10) Our journal encourages inclusion of *data citations in the reference list* to directly cite datasets that were re-used and obtained from public databases. Data citations in the article text are distinct from normal bibliographical citations and should directly link to the database records from which the data can be accessed. In the main text, data citations are formatted as follows: "Data ref: Smith et al, 2001" or "Data ref: NCBI Sequence Read Archive PRJNA342805, 2017". In the Reference list, data citations must be labeled with "[DATASET]". A data reference must provide the database name, accession number/identifiers and a resolvable link to the landing page from which the data can be accessed at the end of the reference. Further instructions are available at .

11) We replaced Supplementary Information with Expanded View (EV) Figures and Tables that are collapsible/expandable online. EV Figures should be cited as 'Figure EV1, Figure EV2' etc... in the text and their respective legends should be included in the main text after the legends of regular figures.

- Additional Tables/Datasets should be labeled and referred to as Table EV1, Dataset EV1, etc. Legends should be provided in a separate tab in case of .xls files. Alternatively, the legend can be supplied as a separate text file (README) and zipped together with the Table/Dataset file.

<https://www.embopress.org/page/journal/17574684/authorguide#expandedview>

12) Author contributions: CRediT has replaced the traditional author contributions section because it offers a systematic machine-readable author contributions format that allows for more effective research assessment. Please remove the Authors Contributions from the manuscript and use the free text boxes beneath each contributing author's name in our system to add specific details on the author's contribution. More information is available in our guide to authors.

13) Disclosure statement and competing interests: We updated our journal's competing interests policy in January 2022 and request authors to consider both actual and perceived competing interests. Please review the policy
<https://www.embopress.org/competing-interests> and update your competing interests if necessary.

14) Every published paper now includes a 'Synopsis' to further enhance discoverability. Synopses are displayed on the journal webpage and are freely accessible to all readers. They include a short stand first (maximum of 300 characters, including space) as well as 2-5 one-sentences bullet points that summarizes the paper. Please write the bullet points to summarize the key NEW findings. They should be designed to be complementary to the abstract - i.e. not repeat the same text. We encourage inclusion of key acronyms and quantitative information (maximum of 30 words / bullet point). Please use the passive voice. Please attach these in a separate file or send them by email, we will incorporate them accordingly.

Please note that these would be the final versions and changes during proofing are usually not allowed.

15) As part of the EMBO Publications transparent editorial process initiative (see our policy here:

https://www.embopress.org/transparent-process#Review_Process), Molecular Systems Biology will publish online a Peer Review File (PRF) to accompany accepted manuscripts.

In the event of acceptance, this file will be published in conjunction with your paper and will include the anonymous referee reports, your point-by-point response and all pertinent correspondence relating to the manuscript. Let us know whether you agree with the publication of the PRF and as here, if you want to remove or not any figures from it prior to publication.

Please note that the Author checklist will be published at the end of the PRF.

Molecular Systems Biology has a "scooping protection" policy, whereby similar findings that are published by others during review or revision are not a criterion for rejection. Should you decide to submit a revised version, I do ask that you get in touch after three months if you have not completed it, to update us on the status.

Yours sincerely,

Reviewer #2:

Authors have addressed all my previous comments. I confirm that the vignette R code works.

Reviewer #3:

The manuscript presents a novel method to infer reversible and irreversible state transitions from snapshot single-cell data. The approach leverages Gene Regulatory Networks (GRNs) to construct vector fields, using cross-cell correlations between the expression levels of regulator genes in one cell and their target genes in neighboring cells. Unlike existing RNA velocity inference methods, this technique does not require spliced and unspliced transcript counts. Overall, the manuscript is well-written and presents promising results.

The authors have satisfactorily address my previous comments. I have no other comments.

Reviewer #4:

I would like to commend the authors for the efforts made in addressing the reviewers comments. While most of the concerns have been addressed either with additional experiments or with an improved discussion, some other concerns remain still (partially) unaddressed.

1) Further ablation-type analyses are needed to justify the methodological choices and their limitations. For example, several reviewers commented on the model's critical reliability on robust and accurate GRNs. The authors have addressed some of these concerns, but on limited examples. It would be very interesting to see how the model's inference changes with different variations of the GRN (besides the limited analysis performed in Appendix Figure S5): e.g., node and edge ablation, and edge permutation. Or when using a bootstrapped GRN from data rather than the true one.

2) Further analysis on GRNs containing more than 30 genes is necessary. While these GRNs may not be highly curated, it would give the reader a sense of the true scalability of the model when using larger GRNs.

3) Additional comparisons against other RNA velocity models should be included such as VeloCycle.

Response letter

We are thankful to the referees for their careful reading of the revised manuscript and for their thoughtful and constructive comments and suggestions. In this second revision, we made sure to address all the issues enumerated by the 4th reviewer. Doing so led to significant improvement of the article and made the descriptions clearer and results sounder. Below we have detailed all the changes made (our response in red in the letter and changes in red in the revised manuscript) point-by-point according to the referees' input (in black).

Reviewer #2:

Authors have addressed all my previous comments. I confirm that the vignette R code works.

Reviewer #3:

The manuscript presents a novel method to infer reversible and irreversible state transitions from snapshot single-cell data. The approach leverages Gene Regulatory Networks (GRNs) to construct vector fields, using cross-cell correlations between the expression levels of regulator genes in one cell and their target genes in neighboring cells. Unlike existing RNA velocity inference methods, this technique does not require spliced and unspliced transcript counts. Overall, the manuscript is well-written and presents promising results.

The authors have satisfactorily address my previous comments. I have no other comments.

We thank the 2nd and 3rd reviewers for their satisfaction with our previous revision.

Reviewer #4:

I would like to commend the authors for the efforts made in addressing the reviewers comments. While most of the concerns have been addressed either with additional experiments or with an improved discussion, some other concerns remain still (partially) unaddressed.

1) Further ablation-type analyses are needed to justify the methodological choices and their limitations. For example, several reviewers commented on the model's critical reliability on robust and accurate GRNs. The authors have addressed some of these concerns, but on limited examples. It would be very interesting to see how the model's inference changes with different variations of the GRN (besides the limited analysis performed in Appendix Figure S5): e.g., node and edge ablation, and edge permutation. Or when using a bootstrapped GRN from data rather than the true one.

We thank the reviewer for their interest in this aspect of our work. In addition to the exhaustive set of node and edge ablations we performed on GRNs in Figs 5, 6, and S5 (now S6), we have

also added an additional figure (Appendix Fig. S7) showing edge ablations for synthetic GRNs (lines 316-322, 333-334). We note that in the larger experimental networks, the majority of minor changes to the network have little effect on the results, while changing key interactions can cause more drastic effects. On the other hand, the small circuit examples demonstrate that when removing an edge substantially changes the predictions, these changes generally appear to be strongest in the region of state space where the edge has the largest effects. As we have applied the approach to several curated, computationally inferred, and synthetic GRNs, we believe that the results we show are sufficient to demonstrate that this analysis can provide insights about a node or edge's importance and role in the GRN.

2) Further analysis on GRNs containing more than 30 genes is necessary. While these GRNs may not be highly curated, it would give the reader a sense of the true scalability of the model when using larger GRNs.

We appreciate the suggestion and have incorporated a more extensive benchmark on GRNs extending up to much larger sizes (Appendix Fig. S9). We note that with respect to computational time, the method is much more sensitive to the number of cells than the size of the GRN. We note that, in the updated appendix figure, we denoted the size of GRNs in terms of edge count rather than node count; these values are correlated, with the maximum node count now at 113. A primary concern with larger GRNs is instead the lack of curation as noted by the reviewer, and a diminished contribution of each individual edge to the predicted vectors, limitations which we discuss in the main text.

3) Additional comparisons against other RNA velocity models should be included such as VeloCycle.

We recognize the need for more thorough comparisons and have compared our results with several different RNA velocity methods (Lines 256-273, Updated Fig. 3, Appendix Fig. S3). While we were unable to obtain results VeloCycle in particular due to its focus on the cell cycle (a dataset for which we did not analyze RNA-seq, but microarray data), we have compared the results of STICCC with a recent method veloVI, as well as with the dynamical model from sevelo, in addition to our prior comparison against sevelo's steady state model. We find overall agreement across the methods, and strong concordance between STICCC and RNA velocity. Comparing the predicted vectors with observed trajectories, we note that STICCC slightly outperforms the other methods. As we note in the revised manuscript, we obtain similar results to these approaches without the use of unspliced counts.

5th Jan 2026

Manuscript Number: MSB-2024-12735RR

Title: Dissecting reversible and irreversible single cell state transitions from gene regulatory networks

Author: Daniel Ramirez

Mingyang Lu

Dear Dr Lu,

Thank you for the submission of your revised manuscript to Molecular Systems Biology. I am pleased to inform you that we will be able to accept your manuscript pending the following final amendments:

- 1) In the main manuscript file, please rename the "Summary" to "Abstract".
- 2) Please reduce the number of keywords to max. 5.
- 3) Please remove the 'Code Availability' section and include the information in the 'Data availability' section formatted according to the example below:

"The datasets and computer code produced in this study are available in the following databases:

- Chip-Seq data: Gene Expression Omnibus GSE46748 (<https://www.ncbi.nlm.nih.gov/geo/query/acc.cgi?acc=GSE46748>)

- Modeling computer scripts: GitHub (<https://github.com/SysBioChalmers/GECKO/releases/tag/v1.0>)

- [data type]: [full name of the resource] [accession number/identifier] ([doi or URL or identifiers.org/DATABASE:ACCESSION])"

- 4) The Data Availability section should only include newly generated code and datasets for the current study. Previously published datasets that have been re-analyzed should be mentioned in the Results/Methods and referenced as a normal Reference or a Data Citation (see below).

- 5) Please update the README files on Github with additional detailed, practical use instructions for potential future users of your code.

- 6) Please rename "Declaration of Interests" to "Disclosure and competing interests statement". We updated our journal's competing interests policy in January 2022 and request authors to consider both actual and perceived competing interests. Please review the policy <https://link.springer.com/partners/embo-press/editorial-policies#Competing%20interest%20disclosures> and update your competing interests if necessary.

- 7) Please correct the reference citation in the reference list to be alphabetical (not numerical). Where there are more than 10 authors on a paper, only the first 10 should be listed, followed by "et al.". Please check "Author Guidelines" for more information: <https://link.springer.com/journal/44320/submission-guidelines#cms-Reference-guidelines>

- 8) Our journal encourages inclusion of *data citations in the reference list* to directly cite datasets that were re-used and obtained from public databases. Data citations in the article text are distinct from normal bibliographical citations and should directly link to the database records from which the data can be accessed. In the main text, data citations are formatted as follows: "Data ref: Smith et al, 2001" or "Data ref: NCBI Sequence Read Archive PRJNA342805, 2017". In the Reference list, data citations must be labeled with "[DATASET]". A data reference must provide the database name, accession number/identifiers and a resolvable link to the landing page from which the data can be accessed at the end of the reference. Further instructions are available at .

- 9) In the Methods, please take care of the following:

- Although you have reported in the Author Checklist that cell lines and microbes were employed in the study, it does not appear as if you actually worked with these experimentally, but rather reused data obtained from the cell lines/microbes. If you did not use them experimentally for the course of the study, please change your answer in the Author Checklist to 'not applicable'. If you did employ them, the Methods section will need to include a section with methodological details such as how the cell lines/microbes were grown/treated, whether or not the cell lines were recently authenticated and tested for mycoplasma contamination, and the information will also need to be included in the Reagents and Tools table.

- 10) All Materials and Methods need to be described in the main text using our 'Structured Methods' format. According to this format, the Methods section includes a Reagents and Tools Table (listing key reagents, experimental models, software and relevant equipment and including their sources and relevant identifiers) followed by a Methods and Protocols section describing the methods, ideally using a step-by-step protocol format. The aim is to facilitate adoption of the methodologies across labs. Please download and fill our Reagents and Tools Table template (.docx), which you can find in our author guidelines: <https://www.embopress.org/page/journal/14693178/authorguide#structuredmethods>.

<https://www.embopress.org/doi/10.15252/msb.20178071>. "

- 11) Please place individual sections of the manuscript in the following order: Title page - Abstract & Keywords - Introduction - Results - Discussion - Methods - Data Availability - Acknowledgements - Disclosure and Competing Interests Statement - References - Figure Legends - Expanded View Figure Legends.

- 12) For the figures and figure legends, please take care of the following:

- Please note that the legend for figure 3 is not provided in the sequential manner. This needs to be rectified.

- Please note that information related to n is missing in the legends of figures 3A, C, E, F, H, J; EV1 F

13) The title page of the Appendix file should contain "Appendix for + manuscript title"; subtitles "Supplementary Figures" and "Supplementary Tables" should be renamed to "Appendix Figures" and "Appendix Tables" respectively; the nomenclature should be Appendix Table S1-S7 throughout the manuscript and Appendix PDF instead of Extended View Table 1-7.

14) As part of the EMBO Publications transparent editorial process initiative (see our policy here:

https://www.embopress.org/transparent-process#Review_Process), Molecular Systems Biology will publish online a Peer Review File (PRF) to accompany accepted manuscripts. This file will be published in conjunction with your paper and will include the anonymous referee reports, your point-by-point response and all pertinent correspondence relating to the manuscript. Let us know whether you agree with the publication of the PRF and as here, if you want to remove or not any figures from it prior to publication. Please note that the Authors checklist will be published at the end of the PRF.

15) After your paper is published, we may promote it on social media. If you have any handles or hashtags for Bluesky you would like included, please let us know.

16) Please provide a point-by-point letter INCLUDING my comments and your detailed responses (as Word file).

I look forward to reading a new revised version of your manuscript as soon as possible.

Yours sincerely,

Poonam Bheda, PhD
Scientific Editor
Molecular Systems Biology

Reviewer #4:

I thanks the authors for addressing my remaining concerns, and I have no other comment or concern.

1) In the main manuscript file, please rename the "Summary" to "Abstract".

We have renamed this section accordingly.

2) Please reduce the number of keywords to max. 5.

We have reduced the keyword list to the 5 most relevant terms.

3) Please remove the 'Code Availability' section and include the information in the 'Data availability' section formatted according to the example below:

"The datasets and computer code produced in this study are available in the following databases:

- Chip-Seq data: Gene Expression Omnibus GSE46748

(<https://www.ncbi.nlm.nih.gov/geo/query/acc.cgi?acc=GSE46748>)

- Modeling computer scripts: GitHub

(<https://github.com/SysBioChalmers/GECKO/releases/tag/v1.0>)

- [data type]: [full name of the resource] [accession number/identifier] ([doi or URL or identifiers.org/DATABASE:ACCESSION])"

We have removed the code availability section and referenced code and datasets produced in this work in the manner described above.

4) The Data Availability section should only include newly generated code and datasets for the current study. Previously published datasets that have been re-analyzed should be mentioned in the Results/Methods and referenced as a normal Reference or a Data Citation (see below).

We have revised the Data Availability section to only include newly generated code and datasets, with re-analyzed data cited with appropriate Data Citations.

5) Please update the README files on Github with additional detailed, practical use instructions for potential future users of your code.

We have updated the README to include more detailed instructions and description of the contents, code examples, and practical tips for using the software.

6) Please rename "Declaration of Interests" to "Disclosure and competing interests statement". We updated our journal's competing interests policy in January 2022 and request authors to consider both actual and perceived competing interests. Please review the policy <https://link.springer.com/partners/embo-press/editorial-policies#Competing%20interest%20disclosures> and update your competing interests if necessary.

We have updated the section title and reviewed the policy and confirmed that there are no competing interests.

7) Please correct the reference citation in the reference list to be alphabetical (not numerical). Where there are more than 10 authors on a paper, only the first 10 should be listed, followed by "et al.". Please check "Author Guidelines" for more information: <https://link.springer.com/journal/44320/submission-guidelines#cms-Reference-guidelines>

We have adjusted the reference style accordingly.

8) Our journal encourages inclusion of *data citations in the reference list* to directly cite datasets that were re-used and obtained from public databases. Data citations in the article text are distinct from normal bibliographical citations and should directly link to the database records from which the data can be accessed. In the main text, data citations are formatted as follows: "Data ref: Smith et al, 2001" or "Data ref: NCBI Sequence Read Archive PRJNA342805, 2017". In the Reference list, data citations must be labeled with "[DATASET]". A data reference must provide the database name, accession number/identifiers and a resolvable link to the landing page from which the data can be accessed at the end of the reference. Further instructions are available at <https://www.embopress.org/page/journal/17574684/authorguide#referencesformat>.

We have added data citations for publicly available data used in our study.

9) In the Methods, please take care of the following:
- Although you have reported in the Author Checklist that cell lines and microbes were employed in the study, it does not appear as if you actually worked with these experimentally, but rather reused data obtained from the cell lines/microbes. If you did not use them experimentally for the course of the study, please change your answer in the Author Checklist to 'not applicable'. If you did employ them, the Methods section will need to include a section with methodological details such as how the cell lines/microbes were grown/treated, whether or not the cell lines were recently authenticated and tested for mycoplasma contamination, and the information will also need to be included in the Reagents and Tools table.

We have amended the Author Checklist to correctly state that no cell lines or microbes were used experimentally in this work.

10) All Materials and Methods need to be described in the main text using our 'Structured Methods' format. According to this format, the Methods section includes a Reagents and Tools Table (listing key reagents, experimental models, software and relevant equipment

and including their sources and relevant identifiers) followed by a Methods and Protocols section describing the methods, ideally using a step-by-step protocol format. The aim is to facilitate adoption of the methodologies across labs.

Please download and fill our Reagents and Tools Table template (.docx), which you can find in our author

guidelines: <https://www.embopress.org/page/journal/14693178/authorguide#structuredmethods>.

An example of a Method paper with Structured Methods can be found here: <https://www.embopress.org/doi/10.15252/msb.20178071>. "

We have attached an additional file containing the Reagents and Tools table based on the provided template.

11) Please place individual sections of the manuscript in the following order: Title page - Abstract & Keywords - Introduction - Results - Discussion - Methods - Data Availability - Acknowledgements - Disclosure and Competing Interests Statement - References - Figure Legends - Expanded View Figure Legends.

We have placed the manuscript sections in the requested order.

12) For the figures and figure legends, please take care of the following:

- Please note that the legend for figure 3 is not provided in the sequential manner. This needs to be rectified.

We thank you for pointing this out and have revised the legend to list the panels in sequential order.

- Please note that information related to n is missing in the legends of figures 3A, C, E, F, H, J; EV1 F

We have modified the figure legends to include n where relevant.

13) The title page of the Appendix file should contain "Appendix for + manuscript title"; subtitles "Supplementary Figures" and "Supplementary Tables" should be renamed to "Appendix Figures" and "Appendix Tables" respectively; the nomenclature should be Appendix Table S1-S7 throughout the manuscript and Appendix PDF instead of Extended View Table 1-7.

We have amended the Appendix file and references throughout the manuscript as advised.

14) As part of the EMBO Publications transparent editorial process initiative (see our policy here: https://www.embopress.org/transparent-process#Review_Process), Molecular Systems Biology will publish online a Peer Review File (PRF) to accompany accepted manuscripts. This file will be published in conjunction with your paper and will include the anonymous referee reports, your point-by-point response and all pertinent correspondence relating to the manuscript. Let us know whether you agree with the publication of the PRF and as here, if you want to remove or not any figures from it prior to publication. Please note that the Authors checklist will be published at the end of the PRF.

We agree to publication of the Peer Review File and have no figures to withhold from publication.

15) After your paper is published, we may promote it on social media. If you have any handles or hashtags for Bluesky you would like included, please let us know.

We have no specific handles or hashtags to include in promotional material.

16) Please provide a point-by-point letter INCLUDING my comments and your detailed responses (as Word file).

26th Jan 2026

Manuscript number: MSB-2024-12735RRR

Title: Dissecting reversible and irreversible single cell state transitions from gene regulatory networks

Dear Dr Lu,

Thank you again for sending us your revised manuscript. We are now satisfied with the modifications made and I am pleased to inform you that your paper has been accepted for publication.

You may qualify for financial assistance for your publication charges - either via a Springer Nature fully open access agreement or an EMBO initiative. Check your eligibility: <https://link.springer.com/journal/44320/how-to-publish-with-us>

Yours sincerely,

Sincerely,

Poonam Bheda, PhD
Scientific Editor
Molecular Systems Biology

>>> Please note that it is Molecular Systems Biology policy for the transcript of the editorial process (containing referee reports and your response letter) to be published as an online supplement to each paper. If you do NOT want this, you will need to inform the Editorial Office via email immediately. More information is available here: <https://link.springer.com/partners/embo-press/editorial-policies#Peer%20review>